

# A 4.5 km resolution Arctic Ocean simulation with the global multi-resolution model FESOM1.4

Qiang Wang[1], Claudia Wekerle[1], Sergey Danilov[1,2], Xuezhu Wang[3,1], and Thomas Jung[1,4]

[1]Alfred Wegener Institute Helmholtz Center for Polar and Marine Research (AWI), Bremerhaven, Germany
[2]Jacobs University Bremen, Bremen, Germany
[3]Hohai University, Nanjing, China
[4]University of Bremen, Bremen, Germany

*Correspondence to:* Qiang Wang (Qiang.Wang@awi.de)

**Abstract.** In the framework of developing a global modeling system which can facilitate modeling studies on Arctic Ocean and high-mid latitude linkage, we evaluate the Arctic Ocean simulated by the multi-resolution ocean sea-ice model FESOM. To explore the value of using high horizontal resolution for Arctic Ocean modeling, we use two global meshes differing in the horizontal resolution only in the Arctic Ocean (24 km vs. 4.5 km). The high resolution significantly improves the model's representation of the Arctic Ocean. The most pronounced improvement is in the Arctic intermediate layer, in terms of both Atlantic Water (AW) mean state and variability. The deepening and thickening bias of the AW layer, a common issue found in coarse resolution simulations, is significantly alleviated by using higher resolution. The topographic steering of the AW is stronger and the seasonal and interannual temperature variability along the ocean bottom topography is enhanced in the high resolution simulation. The high resolution also improves the ocean surface circulation, mainly through a better representation of the narrow straits in the Canadian Arctic Archipelago (CAA). The representation of CAA throughflow not only influences the release of water masses through the other gateways, but also the circulation pathways inside the Arctic Ocean. However, the mean state and variability of Arctic freshwater content and the variability of freshwater transport through the Arctic gateways appear not to be very sensitive to the increase in resolution employed here. We also discuss model issues that are not directly linked to resolution, highlighting the need for further model development including improving parameterizations.

## 1 Introduction

The Arctic Ocean is the smallest among the world oceans, but it is a very important component of the global climate system due to its geographical location. The atmosphere transports moisture to northern high latitudes and supplies freshwater to the land and ocean. By receiving freshwater through river discharge and precipitation the Arctic Ocean is thus a large freshwater reservoir (*Serreze et al.*, 2006; *Dickson et al.*, 2007; *Rudels*, 2015; *Carmack et al.*, 2016). The inflow through Bering Strait is also an Arctic freshwater source because the salinity of Pacific Water is lower than the mean Arctic salinity (*Roach et al.*, 1995; *Woodgate and Aagaard*, 2005). The Arctic Ocean feeds the North Atlantic with its excess freshwater through Fram and Davis Straits (Fig. 1). The released freshwater passes by the deep water formation regions, which could have significant impacts on the large scale ocean circulation (*Aagaard et al.*, 1985; *Goosse et al.*, 1997; *Hakkinen*, 1999; *Holland et al.*, 2001; *Wadley*



*and Bigg*, 2002; *Jungclaus et al.*, 2005; *Arzel et al.*, 2008; *Jahn and Holland*, 2013). The liquid freshwater stored in the upper Arctic Ocean results in a strong stratification and helps to form a permanent halocline. This limits the upward heat flux from the underlying warm water and allows for a persistence of sea ice cover (*Rudels et al.*, 1996). The latter plays a crucial role for the climate by constraining air-sea heat, momentum and constituents exchange. The Arctic Ocean is also fed by warm and saline

Atlantic Water, which circulates mainly cyclonically under the cold halocline and provides a possible heat source of Arctic sea ice basal melting (*Polyakov et al.*, 2010, 2013a). The intermediate Arctic waters leave the Arctic basins through Fram Strait, the only deep Arctic gateway, supplying part of the dense waters that feed the Atlantic overturning circulation (*Rudels and Friedrich*, 2000; *Karcher et al.*, 2011).

Currently the Arctic Ocean is undergoing unprecedented changes, with a freshening trend in the surface layer (*Proshutinsky*

*et al.*, 2009; *McPhee et al.*, 2009; *Rabe et al.*, 2011; *Polyakov et al.*, 2013b; *Haine et al.*, 2015), warming events (*Dmitrenko et al.*, 2008; *Polyakov et al.*, 2012, 2013b), and sea ice decline (*Kwok et al.*, 2009; *Comiso*, 2012; *Cavalieri and Parkinson*, 2012; *Stroeve et al.*, 2012; *Laxon et al.*, 2013). These changes are accompanied not only by a shift in ocean circulation regimes and physical conditions, but also by substantial changes in biogeochemical processes (*Arrigo and van Dijken*, 2015; *Tremblay et al.*, 2015). Through feedbacks with other components of the Earth System, the changes are further accelerated by processes

of Arctic amplification (*Serreze and Barry*, 2011). The on-going and future Arctic changes could have a large influence on lower latitudes ocean and climate with potential social impact, although it is a subject remaining under debate (e.g., *Vihma*, 2014; *Wallace et al.*, 2014).

Numerical modeling can be used to understand the dynamics of the ocean and predict its future changes. Despite a lot of success in modeling studies of the Arctic Ocean, the state-of-the-art ocean general circulation models still show non-negligible

model biases, as illustrated by different model intercomparison studies (*Jahn et al.*, 2012; *Johnson et al.*, 2012; *Aksenov et al.*, 2016; *Wang et al.*, 2016a, b; *Ilicak et al.*, 2016). For example, in the earlier Arctic Ocean Intercomparison Project (AOMIP, *Proshutinsky and Kowalik*, 2007; *Proshutinsky et al.*, 2011), it was identified that too thick Atlantic Water layers in the Arctic Ocean were simulated in the models, very possibly due to spurious numerical mixing (*Holloway et al.*, 2007; *Karcher et al.*, 2007). In the recent Coordinated Ocean-ice Reference Experiments, phase II (CORE-II, *Griffies et al.*, 2012) project, it was

found that this issue still remains one decade later (*Ilicak et al.*, 2016). Large model biases in the upper Arctic Ocean are another common issue in many ocean general circulation models. The mean state of the liquid freshwater and sea ice simulated in the CORE-II models, including their storage and the Arctic-Subarctic fluxes, shows a very pronounced spread among models, although the temporal variability is more consistently represented (*Wang et al.*, 2016a, b). *Wang et al.* (2016b) found that all the CORE-II models experienced a dramatic increase in their Arctic liquid freshwater content during the first few decades

of model simulations. They showed that the spread of simulated liquid freshwater transport through Fram and Davis Straits amounts to as much as $50\%$ of the mean transport values. The large model uncertainty identified in previous studies calls for further model development efforts in the community.

Model simulation results can be sensitive to model configuration, including the choice of numerical and physical schemes, parameters and grid resolution. In this paper we will evaluate a high resolution Arctic Ocean simulation and ellucidate the

sensitivity of model results to resolution. There are many narrow straits in the world's oceans, which play important roles in





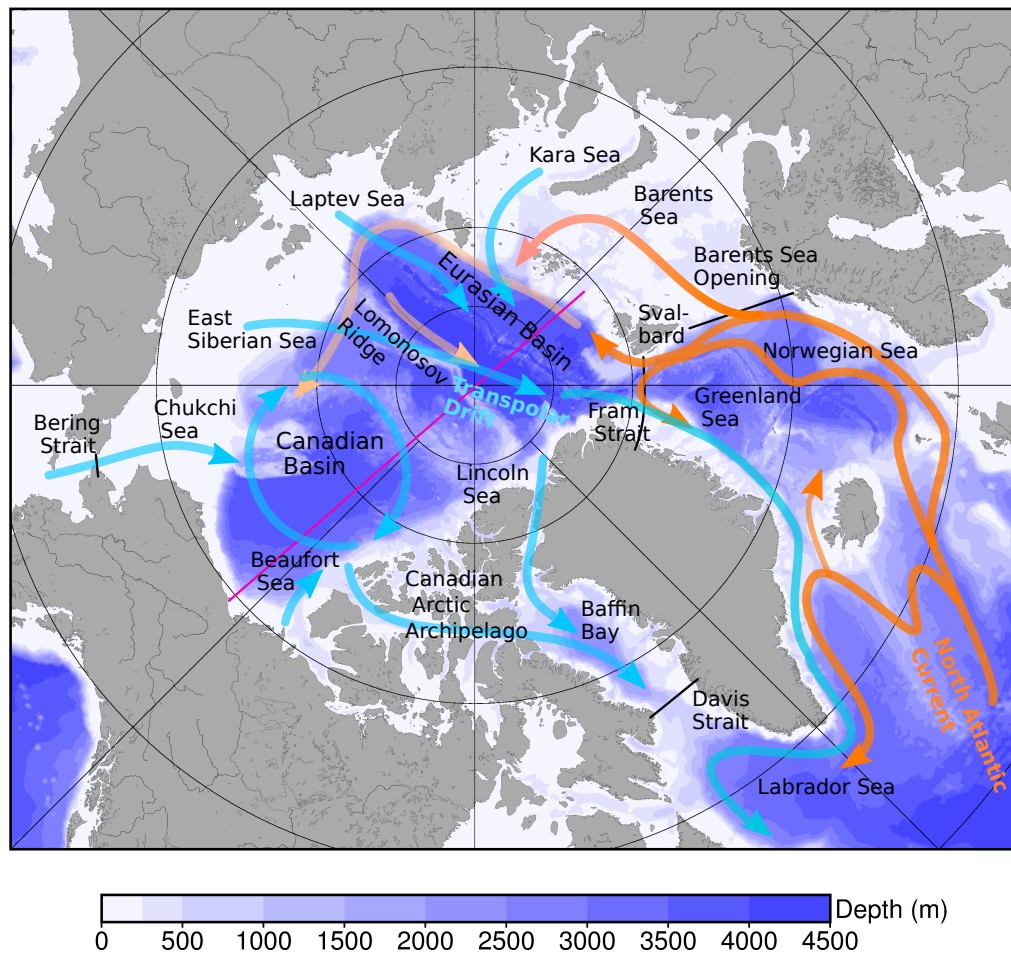

**Figure 1.** Schematic of main ocean circulations in the pan-Arctic Ocean. The freshwater circulation is shown with light blue curves, and the Atlantic Water (AW) circulation is shown with orange curves. The color patch in the background shows the ocean bottom bathymetry. The black lines indicate the four Arctic gateways. The magenta line crossing the North Pole indicates the location of the transect shown in Fig. 8e,f.





connecting different ocean basins but are difficult to resolve with resolution typically used in large scale ocean models. The Arctic Ocean is enclosed by continents and connected to lower latitude oceans via narrow straits. Especially, the three main Canadian Arctic Archipelago (CAA) channels have widths of about 10 km, 30 km and 50 km at their narrowest locations (*Melling*, 2000). It was shown that the ocean fluxes through these narrow channels can be reasonably resolved when using

very high horizontal resolution (about 4 km, *Wekerle et al.*, 2013). High resolution is also required to resolve small scale dynamics which could have impact on larger scale circulation and water mass properties. As the first baroclinic Rossby radius is very small in the Arctic Ocean (*Nurser and Bacon*, 2014), mesoscale-eddy resolving is difficult to achieve even in regional Arctic Ocean models. For process studies, however, simulations with 1–2 km resolution have been used to resolve mesoscale dynamics and ocean circulation in Fram Strait (*Kawasaki and Hasumi*, 2015; *Hattermann et al.*, 2016). As computational

resources grow with time, the modeling community tends to use higher and higher model resolution. Certainly there is a need in the modeling community to evaluate high resolution models with respect to the common model issues identified in previous model studies.

In the framework of our own model development, we aim to develop a coupled model system that can facilitate to carry out climate research with focus on the Arctic Ocean and Arctic lower-latitude linkage. We use the global Finite Element Sea

ice Ocean Model (FESOM, *Wang et al.*, 2014) as its ocean/sea ice component. This model employs unstructured meshes and allows for variable resolution without traditional nesting. With it we are able to allocate finer resolution in the northern high latitudes than in many other parts of the global ocean. However, practically optimal ocean resolution in the Arctic Ocean needs to be decided. The finally chosen resolution should help to adequately simulate the key ocean dynamics with confined model biases. At the same time the model system should not be too costly as it will be used in long climate simulations. As one of

the first steps towards designing such a system, in this paper we evaluate the simulated Arctic Ocean by FESOM on a global mesh with 4.5 km resolution inside the Arctic Ocean. This resolution is still not mesoscale eddy resolving in the large part of the Arctic Ocean (see Fig. 2). We use it in the current stage because it is already much higher than typical resolutions used in current climate models (one fourth to one degree) while we still obtain a reasonably high model throughput of about 7 model years per day.

For our purpose we carried out ocean simulations driven by prescribed CORE-II atmospheric forcing. A coarse resolution setup of FESOM (with 24 km in the Arctic Ocean) has been used in previous CORE-II studies. We will compare the 4.5 km model results with those from this coarse setup to understand the impace of model resolution. The most climate-relevant metrics of the Arctic Ocean, that is, the Atlantic Water property, the Arctic freshwater budget and sea ice state were used to evaluate the state-of-the-art ocean climate models in the CORE-II Arctic studies (*Ilicak et al.*, 2016; *Wang et al.*, 2016a, b).

These studies provided background knowledge on the Arctic Ocean representation in those models and identified their common issues. In this paper we will mainly focus on the key diagnostics used in these studies for evaluating our simulations.

The model setups will be described in Section 2. The results about Atlantic Water and freshwater budget of the Arctic Ocean will be presented in Sections 3 and 4, respectively, followed by discussions (Section 5) and summary (Section 6).





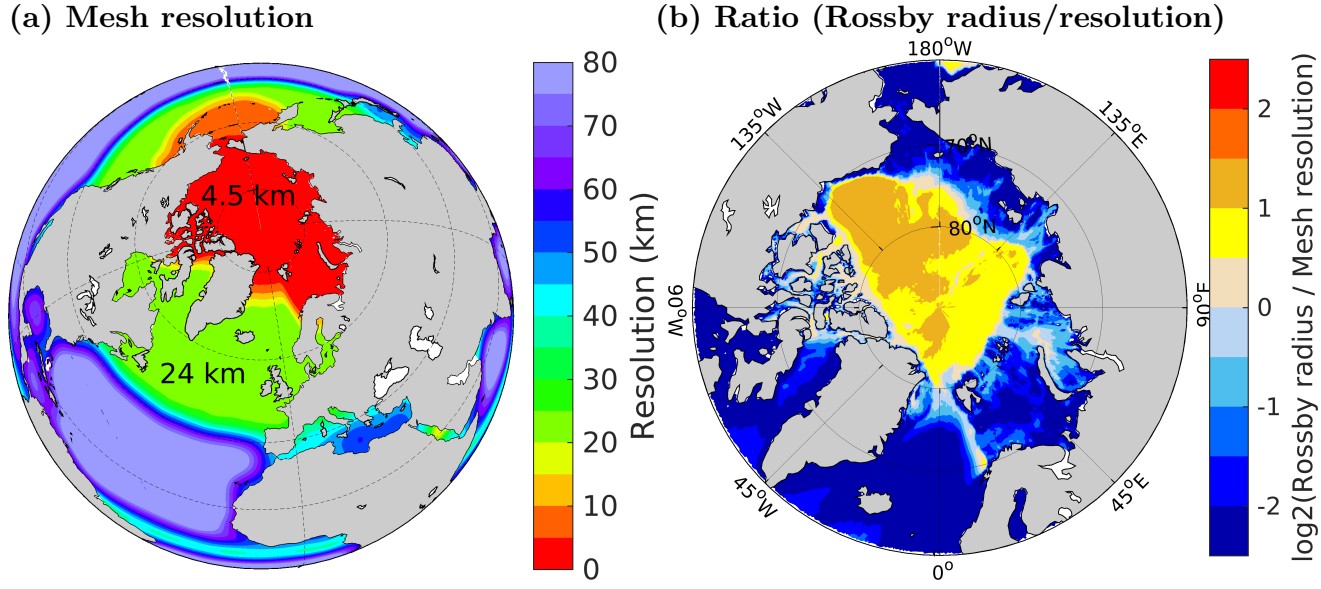

**Figure 2.** (a) The horizontal grid size of mesh HIGH (4.5 km in the Arctic Ocean). (b)Ratio between the first baroclinic Rossby radius and grid size shown with the log2 scale. Values larger than one indicate mesoscale eddy resolving. Note that effective model resolution usually is coarser than the grid size due to numerical dissipation. The Rossby radius is calculated for each season and the local minimum is used for (b).

## 2 Model setup

The latest version of FESOM (*Wang et al.*, 2014) is used in this study. The ocean dynamical core stems from the early study of *Danilov et al.* (2004) and *Wang et al.* (2008). It work with unstructured triangular meshes, so variable grid resolution can be conveniently applied without the necessity of using traditional nesting. It is coupled to a dynamic-thermodynamic sea ice model

5   (*Timmermann et al.*, 2009; *Danilov et al.*, 2015), which is based on the *Parkinson and Washington* (1979) thermodynamics and uses an updated version of the elastic-viscos-plastic (EVP, *Hunke and Dukowicz*, 1997) rheology. The sea ice model is discretized on the same surface mesh as the ocean model by using an unstructured-mesh method too.

A blend of two bottom topography data sets is used. North of 69°N the 2 km resolution version of the International Bathymetric Chart of the Arctic Oceans (IBCAO, *Jakobsson et al.*, 2008) is used, while south of 64°N the 1 min resolution version

10   of the General Bathymetric Chart of the Oceans (GEBCO) is used. Between 64°N and 69°N the topography is taken as a linear combination of the two data sets. An explicit second-order flux-corrected-transport (FCT) scheme (*Löhner et al.*, 1987) is employed in the tracer equations. It helps to preserve monotonicity and eliminate overshoots. When compared to a second order scheme without flux limiter and an implicit second order scheme in idealized 2-D test cases, at coarse resolution this FCT scheme tends to slightly reduce local maxima even for a smooth field, but at high resolution it well represents sharp fronts

15   and shows least dispersion errors (*Wang*, 2007).





The diapycnal mixing is parameterized with the k-profile scheme proposed by *Large et al.* (1994). In case of static instability the vertical mixing coefficients are increased as a parametrization for unresolved vertical overturning processes. We apply biharmonic friction with a *Smagorinsky* (1963) viscosity, which is flow-dependent. The *Redi* (1982) isoneutral diffusion with small slope approximation and the *Gent and McWilliams* (1990, GM) parameterization in a skew diffusion form (*Griffies*, 1998)

are used. A reference value is determined for neutral diffusivity and GM thickness diffusivity at each surface grid location by considering the local horizontal resolution. It is then scaled by the squared buoyancy frequency to obtain 3D diffusivity (*Wang et al.*, 2014).

Two global meshes are compared in this study. The first one (LOW) has 1 degree nominal horizontal resolution in most part of the world's ocean. In the equatorial band the resolution is doubled, and north of $45^\circ$N the resolution is set to about

24 km. On the second mesh (HIGH) the horizontal resolution is further increased to 4.5 km inside the Arctic Ocean (defined by the Arctic gateways of Bering Strait, CAA, Fram Strait and Barents Sea Opening, Fig. 2a). Mesh LOW has been used in the CORE-II model intercomparison studies, and mesh HIGH has been used in a recent study on Arctic sea ice leads (*Wang et al.*, 2016c). Judged by comparing the Rossby radius and grid size without considering numerical dissipation, mesh HIGH is not eddy permitting over the continental shelves and in Fram Strait, eddy permitting in the Eurasian Basin, and partly eddy

resolving in the Canadian Basin (Fig. 2b). The reference value of the neutral and GM thickness diffusivity is $50\,\mathrm{m}^2/\mathrm{s}$ at 24 km resolution, and $4\,\mathrm{m}^2/\mathrm{s}$ at 4.5 km. In the vertical 47 z-levels are used with resolution of 10 m in the top 100 m and gradually decreased downwards.

The high resolution simulation allows for a model throughput of 7 model years per day, which is about one third of the low resolution simulation. Three passive tracers are used to illustrate the pathways of different inflow water masses from Fram

Strait, Barents Sea Opening (BSO) and Bering Strait. Initially the concentration of these tracers are set to zero. During the simulations they are restored to one over the ocean column within these straits. Adding these passive tracers reduces the model throughput by about 20%.

As discussed by *Griffies et al.* (2009), ocean climate models without a coupled active atmospheric model lack many of the feedbacks present in a fully coupled system, which necessitates restoring of model sea surface salinity (SSS) to observed

climatological SSS in global ocean-sea ice models. In addition, SSS restoring helps to avoid unbounded local salinity trends that can occur in response to inaccuracies in, for example, precipitation. The strength of SSS restoring (defined by a piston velocity) in our simulations is $50\,\mathrm{m}$ over $300\,\mathrm{days}$, a value used in many CORE-II models (*Danabasoglu et al.*, 2014). The impact of SSS restoring will be discussed at the end in the discussion section.

The model is forced by the CORE-II interannual atmospheric data set (*Large and Yeager*, 2009) from 1950 to 2009. The

ocean is initialized with the temperature and salinity fields from the Polar Science Center Hydrographic Climatology v.3 (PHC3, *Steele et al.*, 2001) and starts from a steady state, and sea ice is initialized with climatological fields obtained from a previous simulation. Interannual monthly mean river runoff is taken from the data provided by *Dai et al.* (2009), and in the model the river water is spread over a range of 300 km near river mouths to count for unresolved processes (*Wang et al.*, 2014).





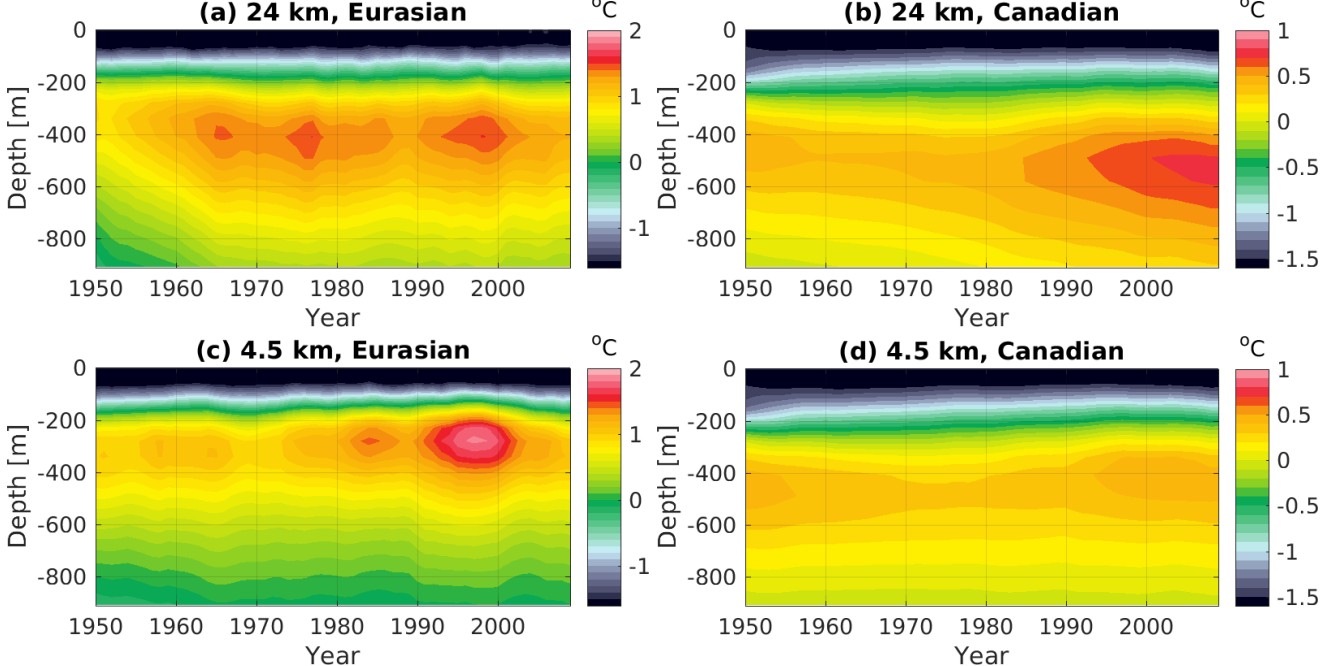

**Figure 3.** Hovmöller diagram of mean potential temperature for (a) Eurasian Basin and (b) Canadian Basin obtained in the simulation LOW. (c),(d) are the same as (a),(b) but for the simulation HIGH.

# 3 Atlantic Water in the Arctic Ocean

At the beginning of this and the next sections we will briefly introduce the present understanding of the Arctic Ocean dynamics and changes, and the major issues to be discussed.

## 3.1 Background

5 A schematic of Atlantic Water (AW) circulation in the pan-Arctic Ocean is shown in Fig. 1. Saline and warm AW enters the Nordic Seas via the northern limb of the North Atlantic Current through the Greenland-Scotland Ridge, and continues northwards in the Nordic Seas in two branches of the Norwegian Atlantic Current (NwAC, *Orvik and Niiler*, 2002). When approaching the BSO, the eastern branch bifurcates with one branch entering the shallow Barents Sea and the other flowing towards the Fram Strait. The AW that enters the Barents Sea looses most of its heat (*Skagseth et al.*, 2008; *Smedsrud et al.*,

10 2013), and these modified waters flow into the intermediate layer of the Arctic Ocean via St. Anna Trough, or contribute to the halocline (*Karcher and Oberhuber*, 2002; *Dmitrenko et al.*, 2011, 2015; *Aksenov et al.*, 2011). The western branch of the NwAC and the remainder of the eastern NwAC branch continue towards the Fram Strait, and form the West Spitsbergen Current (WSC). At Fram Strait, a fraction of AW carried in the WSC recirculates and flows southwards in the East Greenland





Current. The remaining part of the WSC enters the Arctic Ocean at depth, carrying the heat of the AW (*Rudels and Friedrich*, 2000; *Schauer et al.*, 2008; *Beszczynska-Moeller et al.*, 2012).

The AW below the halocline circulates mainly cyclonically along the continental slope and mid-ocean ridges as topographically steered boundary currents (*Rudels et al.*, 1994; *Karcher et al.*, 2003). The warmer Fram Strait branch and colder BSO

branch converge north of the Kara Sea (*Schauer et al.*, 2002; *Karcher and Oberhuber*, 2002; *Maslowski et al.*, 2004) and continue eastward along the Eurasian slope. After passing the Laptev Sea slope, the boundary current bifurcates into one branch following the Lomonosov Ridge and another following the continental slope (*Woodgate et al.*, 2001). The former brings the AW toward Fram Strait, while the latter continues into the Canadian Basin. Interannual changes in AW temperature can propagate into the Arctic Ocean via the Fram Strait inflow, leading to temperature variability along the AW boundary current (*Gerdes*

*et al.*, 2003). Pronounced warming events in the Arctic AW layer have been observed in recent decades (*Polyakov et al.*, 2012, 2013b). These recent unprecedented warming implies that the Arctic deep basins are undergoing significant changes.

In previous model intercomparison studies with focus on Arctic AW (*Holloway et al.*, 2007; *Karcher et al.*, 2007), it was found that one outstanding issue in most ocean models is the unrealistic deepening and thickening of the AW layer. Numerical mixing associated with the advection operator was suggested to be the major cause (*Holloway et al.*, 2007). The recent CORE-

II study indicates that the state-of-the-art ocean general circulation models which are currently used in climate studies still suffer from the deepening of the AW layer (*Ilicak et al.*, 2016). In the following we will explore whether and to what extent this problem can be alleviated by increasing horizontal resolution. Besides the mean state of the AW, we will investigate the model representation of decadal warming and variability on seasonal and interannual time scales.

### 3.2 Spin-up of the AW in Arctic basins

The annual mean temperature horizontally averaged in Eurasian and Canadian basins is plotted as a function of time and depth in Fig. 3. The basin mean temperature shows a very different temporal evolution in the two simulations. In the Eurasian Basin the warm AW layer thickens with time during the first 15 model years in the low resolution simulation (LOW), while the layer thickness remains quasi-steady (up to interannual variability) in the high resolution simulation (HIGH). After initial spin-up the depth of temperature maxima is located at about 400 m in LOW, while in HIGH it remains at about 300 m, the observed

depth suggested by the hydrographic climatology. In the Canadian Basin the thickening and deepening of the AW layer is also very obvious in LOW. In this simulation the core of the AW layer deepens by about 100 m, changing from about 450 m to 550 m during the 60 model years. The model drift in the AW layer occurring during model spin-up is irreversible afterwards. The longer simulation presented in the CORE-II model intercomparison work indicates that the depth of the AW layer temperature maxima in the Canadian Basin continues deepening and stays at around 600 m depth after 300 model years (*Ilicak et al.*, 2016).

In HIGH no thickening and deepening trend is found in the Canadian Basin. In both simulations the Eurasian Basin is featured with decadal warming events, and the Canadian Basin shows more pronounced warming in recent years. Besides the mean state, the two simulations are also different in their representation of variability and decadal changes, which will be assessed below.



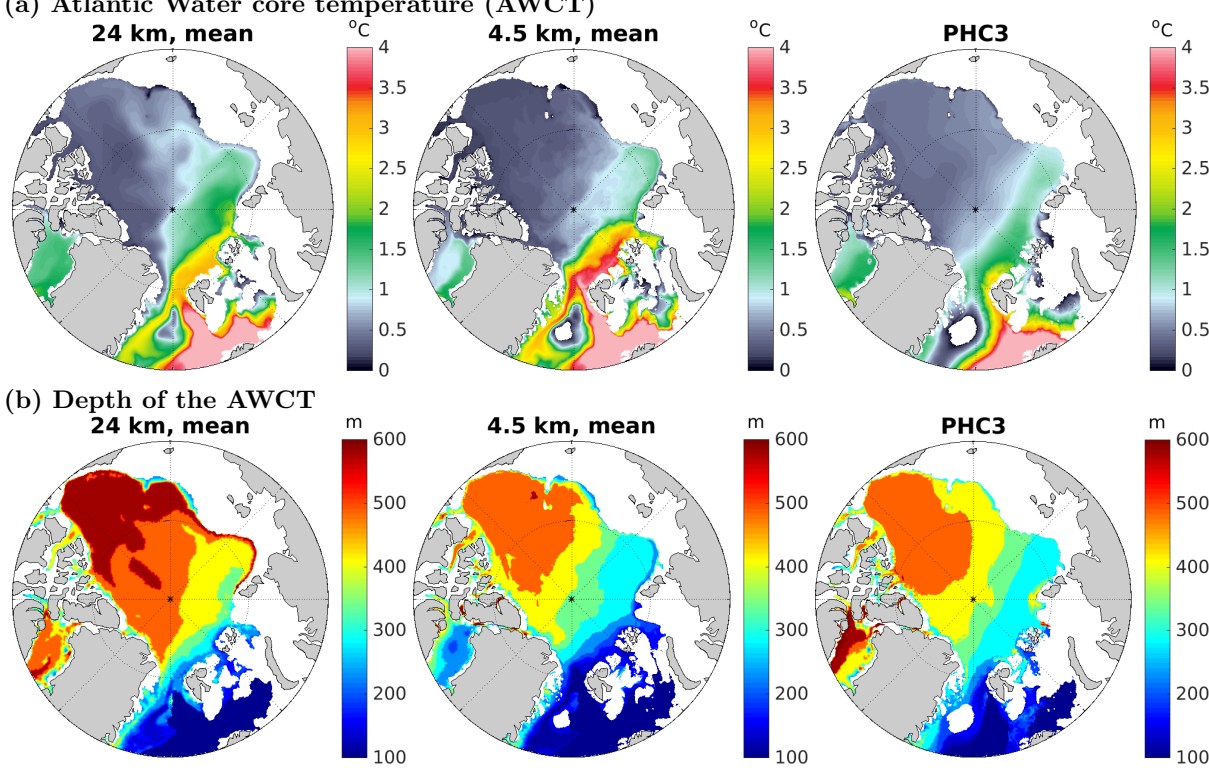

**Figure 4.** (a) Atlantic Water core temperature (AWCT) for (from left to right) simulations LOW and HIGH and the PHC climatology. (b) The same as (a) but for the depth of AWCT. Shelf regions (<200 m) are not shown. The model results are averaged from 1970 to 1999.

### 3.3 Mean state of AW

To assess the spatial distribution of the AW in the Arctic Ocean, we show the Atlantic Water core temperature (AWCT) derived from 30 years mean model results in Fig. 4a. The AWCT is defined as the maximum temperature over the depth at each location. The typical spatial pattern of AW is shown by the climatology. The WSC brings warm AW into Fram Strait, with a fraction recirculating southwards and the remaining part entering the Arctic Ocean. The latter passes the northern slope of Svalbard and flows along the continental slope eastward in the Eurasian Basin. There is a strong contrast in temperature between the Eurasian and Canadian Basins, separated by the Lomonosov Ridge. The cold Barents Sea branch of AW enters the basin at St. Anna Trough and circulates cyclonically as boundary current over the continental slope. Although both simulations can capture these main features, the warm AW is more confined in the Eurasian Basin in simulation HIGH than LOW. The AW boundary current starting from St. Anna Trough towards Lomonosov Ridge is much narrower in simulation HIGH, while it is horizontally more spread in LOW. The observed AWCT is located above 300 m depth in most part of the Eurasian Basin, and deepens towards the Beaufort Sea (Fig. 4b). Simulation HIGH largely reproduces the spatial change of the AWCT depth,





**Figure 5.** (a) Mean temperature profiles in the Eurasian and Canadian Basins in the low and high resolution simulations. (b) The same as (a) but for salinity. (c) The same as (b) but for comparing two low resolution simulations with and without sea surface salinity (SSS) restoring. Model results are averaged from 1980 to 1999.

and the depth in both the Eurasian and Canadian Basins is well represented. In simulation LOW the AWCT is deeper than the observation in most of the Arctic regions, and the contrast between the Eurasian and Canadian Basins is not as obvious as in the observation and simulation HIGH.

Simulation LOW obtains a vertically extended AW layer on basin scales, as shown by the mean temperature profiles in the two basins (Fig. 5a). In this simulation the depth of temperature maxima deepens by about 100 m in both Arctic basins, with the vertical extent of the warm layer reaching much deeper depth. The maximum temperature in Eurasian Basin in simulation



HIGH is about $0.4^{\circ}$C higher than that in the PHC3 data, but the observed depth of the temperature maxima, at about 300 m, is captured by this simulation. In the Canadian Basin the temperature of the Pacific Winter Water (located between the Pacific Summer Water and about 200 m depth) is overestimated in both simulations, implying too strong vertical diffusion, which mixes the cold water with warmer AW below. This feature is obviously not linked to model horizontal resolution and will be discussed in Section 5.

The AW circulation pattern was examined by comparing the topostrophy in the two simulations. Cyclonic circulation dominates the AW layer boundary currents in both ocean basins similarly in the two simulations (not shown). The fact that simulation LOW also has the correct circulation direction in Canadian Basin is very possibly just because its resolution is already fine compared to those with problems reported in previous studies (e.g., *Holloway et al.*, 2007). Indeed, the Arctic Ocean hydrography obtained on mesh LOW was found to be one of the well simulated when comparing the state-of-the-art ocean climate models (*Ilicak et al.*, 2016).

### 3.4 Variability of AW

Although the Arctic Ocean is at the far end of the North Atlantic Current northern limb, strong warming events have been observed in the Arctic AW layer at the end of the 20th century and beginning of the 21st century (*Gerdes et al.*, 2003; *Karcher et al.*, 2003; *Polyakov et al.*, 2012). Despite very limited observations in the remote Arctic deep basins, averaged over decadal time scales the warming events are outstanding and the compiled datasets can be used to assess the model representation of the AW warming (*Polyakov et al.*, 2012). In Fig. 6a the warming in the 1990s relative to the 1970s in the two model simulations and observation is shown. The observation indicates a basinwise warming by about $1^{\circ}$C in the Eurasian Basin, which propagates into the Canadian Basin crossing the Lomonosov Ridge along the continental slope. Simulation HIGH similarly shows a basinwise warming in the Eurasian Basin, and slightly weaker penetration of the warming signal into the Canadian Basin compared to the observation. In this simulation the boundary current along the continental slope and Lomonosov Ridge shows stronger warming than the basin interior, which is not seen in the observation. This could be partly due to the sparseness of hydrography observations. Simulation LOW also obtains a warming signal in the 1990s, but mainly in the eastern Eurasian Basin and over a large part of the Canadian Basin. As shown by the time series of the basin mean temperature in Fig. 3, there is a strong warming and deepening trend in the Canadian Basin throughout simulation LOW. This model drift can explain part of the warming in LOW shown in Fig.6a .

The depth of the AWCT became shallower in the 1990s compared to the 1970s in the observation (Fig. 6b). Simulation HIGH shows a consistent pattern in the change of AWCT depth, while the magnitude is about half of the observed. In simulation LOW the depth becomes shallower in a small region in the sector of the Siberian and Chukchi Seas, but it becomes deeper by about 100 m north of the CAA. The latter can be attributed to the deepening trend of the AW in the model as shown in Fig. 3.

In the following we will focus on the resolution dependency of the interannual variability of AWCT in the two simulations. We use the standard deviation (std) of annual mean AWCT as an indicator of the interannual variability. As shown in Fig. 7, the interannual variability is stronger in the Eurasian Basin and weakens along the AW advection pathway in both simulations. In simulation HIGH, the std is more than $0.4^{\circ}$C in front of the Eurasian continental slope and along the Lomonosov Ridge toward





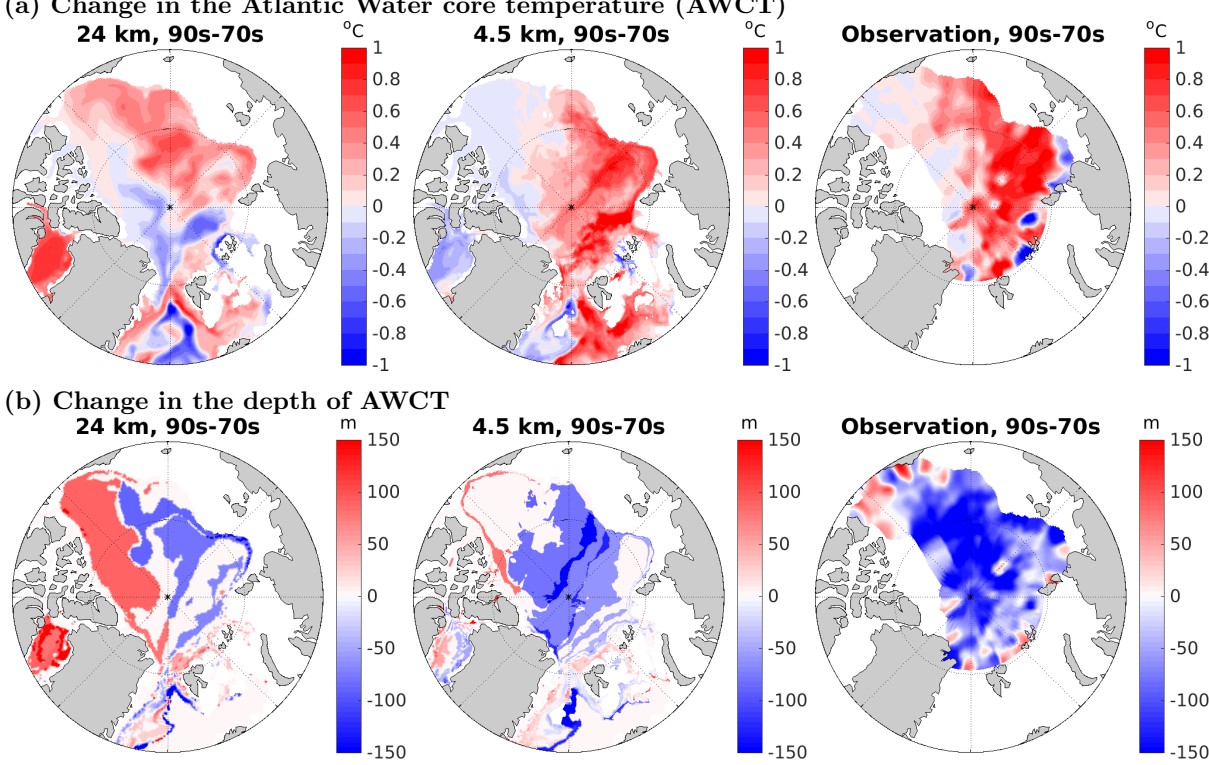

**Figure 6.** (a) Difference of the Atlantic Water core temperature (AWCT) between the 1990s mean and 1970s mean (the former minus the latter) in simulations (from left to right) LOW and HIGH and in observation (*Polyakov et al.*, 2012). (b) The same as (a) but for the depth of the AWCT. For the model results the shelf regions (<200 m) are not shown.

the North Pole. The highest std is found in the western Eurasian Basin where the boundary of the inflowing AW on the interior side changes its location most significantly. In simulation LOW, the std is in the range of $0.2 - 0.3^{\circ}$C along the path of the AW circulation. Different from HIGH, there is no clear indication of stronger interannual variability along the topographically steered boundary current in LOW. The interannual variability is advected to a larger area in the Canadian Basin in LOW, which

5   is consistent to the larger horizontal spreading of AW (Fig. 4a). Using a different model with resolution of 10 km, *Lique and Steele* (2012) showed that the std of AWCT is in a range of about $0.1 - 0.4^{\circ}$C in the Eurasian Basin, similar to that in our simulations. However, the spatial pattern of the std is very different from any of our simulations. In their simulation the highest interannual variability is found starting from the Laptev Sea coast toward the Lomonosov Ridge directly crossing the Eurasian Basin (figure 13 of *Lique and Steele*, 2012). In this respect the difference in the AW interannual variability induced by different

10  resolutions, although significant, is less pronounced than the difference between two models.

The magnitude of mean seasonal cycle of the AWCT is also compared in Fig. 7. Both simulations show that the seasonal variability is advected from Fram Strait into the Arctic interior along the AW boundary current, and then the variability is




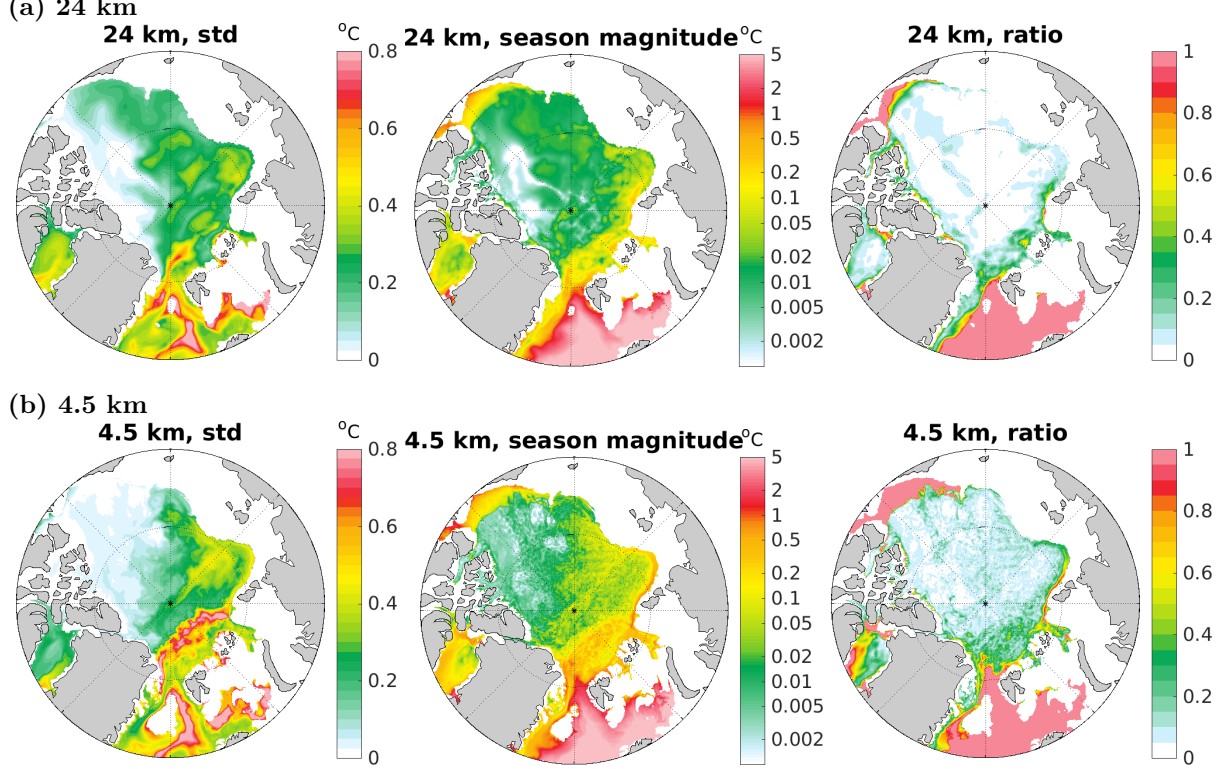

**Figure 7.** (left) The standard deviation (std) of the annual mean Atlantic Water core temperature (AWCT), (middle) the magnitude of the AWCT mean seasonal variability, and (right) the ratio between the seasonal magnitude and the std of the annual mean, in (a) simulation LOW and (b) HIGH. Note that nonlinear color scales are used in the plots of seasonal variability.

re-energized at St. Anna Trough by the BSO branch. In simulation HIGH the magnitude of the AWCT seasonal cycle is nearly $0.5^{\circ}$C in the boundary current downstream St. Anna Trough, and decreases to about $0.2^{\circ}$C over the Laptev Sea continental slope. In simulation LOW the magnitude of the seasonal variability along the continental slope is about half of that in HIGH. When the boundary current bifurcates, with one branch circulating northward along the Lomonosov Ridge and another pene-
trating into the Canadian Basin, the seasonal variability also propagates further along these branches. However, the magnitude becomes smaller with distance, which is less than $0.05^{\circ}$C in HIGH and even lower in LOW. When compared to the strength of interannual variability, in both simulations the seasonal variability is negligible except in the region north of Svalbard and within the narrow boundary current. The most significant seasonal variability is found within the narrow boundary current between St. Anna Trough and the Laptev Sea continental slope in simulation HIGH, where the ratio between the magnitude of the AWCT seasonal cycle and the std of the annual mean AWCT is about 0.8. Therefore, the BSO branch supplies a large part of the seasonal variability shown in this slope region. The spatial pattern of the AWCT seasonal variability in the Arctic Ocean in our simulations is similar to that derived from a different model at 10 km resolution shown by *Lique and Steele* (2012).





However, the strength of the AWCT seasonal variability in their model simulation is similar to that obtained in our simulation LOW and lower than in HIGH.

## 4 Salinity and freshwater budget

### 4.1 Background

A schematic of freshwater (FW) circulation in the pan-Arctic Ocean is shown in Fig. 1. The Arctic Ocean receives a large amount of FW from river runoff, net precipitation, and Pacific Water through Bering Strait (*Serreze et al.*, 2006; *Dickson et al.*, 2007; *Haine et al.*, 2015; *Carmack et al.*, 2016). Liquid FW is stored in the Arctic Ocean with a very non-uniform spatial distribution. The Canadian Basin is characterized by the largest FW content (defined as the amount of pure FW that could be taken out of the upper ocean so that the ocean salinity is changed to 34.8, the Arctic Ocean reference salinity (*Aagaard and Carmack*,

1989)). Especially in the Beaufort Gyre the FW amounts to about 20 m, while it is about 5-10 m in the Eurasian Basin (e.g., *Rabe et al.*, 2011). The anticyclonic Beaufort Gyre is driven by the Beaufort Sea High in atmospheric pressure, which changes the FW content in Beaufort Gyre and the FW distribution between the ocean basins by modulating convergence/divergence of Ekman transport (e.g., *Proshutinsky et al.*, 2002, 2009; *Giles et al.*, 2012). Wind variability over continental shelves can locally induce more significant changes in FW content than the variability from river fluxes, and the variation in large scale

atmospheric circulation (Arctic Oscillation) can modify the pathway of river runoff, thus changing the FW distribution between the Arctic basins (*Dmitrenko et al.*, 2008; *Morison et al.*, 2012).

Both liquid FW and sea ice are drained by the Transpolar Drift and released through Fram Strait. The liquid FW exported through Fram Strait is slightly larger than sea ice export (*Serreze et al.*, 2006), but the difference has increased during the last decade (*Haine et al.*, 2015). Arctic liquid FW is also released to lower latitudes through the CAA and then Davis Strait, with

an amount similar to that released through Fram Strait (*Serreze et al.*, 2006; *Curry et al.*, 2014). Sea ice export through Davis Strait is much less than that from Fram Strait. The possible climate relevance of the FW cycle in the Arctic Ocean and FW release to the North Atlantic is one of the main reasons for continued research on the Arctic FW budget in both the observation and modeling communities (*Carmack et al.*, 2016).

The liquid FW stored in the Arctic Ocean has been increasing starting from the mid-1990s as shown by observations

(*Proshutinsky et al.*, 2009; *McPhee et al.*, 2009; *Giles et al.*, 2012; *Polyakov et al.*, 2013b; *Rabe et al.*, 2014), while sea ice has a persistent declining trend in thickness and volume (*Kwok et al.*, 2009; *Laxon et al.*, 2013). The liquid FW export through Davis Strait was observed to be lower in the 2000s than in the 1990s (*Curry et al.*, 2014), while the Fram Strait liquid FW export has slightly increased in the 2000s compared to the climatological value (*Haine et al.*, 2015). In recent CORE-II model studies using a suite of global ocean-ice models, it was shown that the recent increase in Arctic liquid FW content is

caused by both sea ice melting and reduction of total liquid FW export, with the former being more significant in most of the models (*Wang et al.*, 2016b). However, current observations, especially those of liquid FW budget, are still too sparse for the purpose of quantitative verification of the finding based on models.


**Figure 8.** Hovmöller diagram of mean salinity for (a) the Eurasian Basin and (b) the Canadian Basin obtained in simulation LOW. (c),(d) are the same as (a),(b) but for simulation HIGH. (e) Salinity at the transect along the $140^{\circ}$W$/40^{\circ}$E longitude averaged over the 1980-1999 period in simulation HIGH, shown by the color patch and solid contours. The dashed contours indicate the salinity at the beginning of the model simulation. (f) The same as (e) but for the passive tracer released in Bering Strait. The location of the transect is indicated in Fig. 1.





In the CORE-II model intercomparison project, it was found that the simulated mean state of Arctic FW (FW content and its spatial distribution, and FW transport through Arctic gateways) has significantly large model spreads, even though the same atmospheric forcing was used (*Wang et al.*, 2016a, b). Interannual variability of FW export via Fram Strait is the least consistently simulated among different Arctic gateways, in both AOMIP and CORE-II models (*Jahn et al.*, 2012; *Wang et al.*, 2016b). Besides, all the CORE-II models show a dramatic increase in the simulated Arctic liquid FW content during the model spin-up phase; afterwards, the FW content stays at the overestimated level (unless overestimated AW salt inflow causes it to drop in one particular model, *Wang et al.*, 2016b). There is indication in some studies that higher model resolution might improve the pathway and spatial distribution of liquid FW (*Koldunov et al.*, 2014; *Aksenov et al.*, 2016). In the following we will compare the Arctic FW budget simulated with FESOM using two different horizontal resolutions. The focus will be on the impact on model spin-up, mean state, and interannual to decadal variability of the FW budget.

## 4.2  Spin-up of salinity and freshwater content

The annual mean salinity horizontally averaged over Eurasian and Canadian basins is plotted as a function of time and depth in Fig. 8a-d. In both basins the salinity decreases with time, and it takes nearly 30 years for salinity to spin up to a quasi-equilibrium state in both basins. The two simulations show very similar results, except that the salinity drift in the Eurasian Basin takes place in a relatively shorter period (about 20 years) in the high resolution (HIGH) than in the low resolution simulation (LOW). In Eurasian Basin the salinity drift takes place mainly in the upper 200 m, while in the Canadian Basin mainly between 100 and 400 m depth. The different behavior implies that processes associated with the salinity drifts are different in the two basins.

The freshening of the Eurasian Basin in HIGH is illustrated in a transect along the $140^{\circ}$W/$40^{\circ}$E longitude line in Fig. 8e. Compared to the mean salinity in the first model year, the salinity becomes considerably lower near the Lomonosov Ridge (located near the North Pole in this transect) after the model spin-up phase. The location of strong freshening coincides with the pathway of Pacific Water through Bering Strait (Fig. 8f), which is carried by the Transpolar Drift together with FW from Eurasian riverine. Therefore, the freshening of the Eurasian Basin could be linked to model representation of the upper ocean circulation pathway and the spatial distribution of FW from Bering Strait and river runoff. In simulation LOW we obtain similar results, so changing model resolution does not influence the occurrence of this salinity drift.

The salinity drift is manifested in the time series of Arctic Ocean FW content (Fig. 9). In both simulations the total Arctic liquid FW content increases nearly linearly in the first 20 years. The increase takes place mainly in the two basins, with a similar magnitude. In the Canadian Basin the FW contents are almost identical in the two simulations for all the time, while the FW content in the Eurasian Basin is about 20% higher in simulation LOW after 30 model years. To explain the latter we carried out one sensitivity experiment. We use a model grid similar to LOW in most parts of the global ocean, but increase the resolution to 4.5 km only inside the CAA straits. This mesh has been used in the CAA throughflow study by *Wekerle et al.* (2013). The spatial patterns of mean FW content (in m) from the three simulations are shown in Fig. 10a-c. The sensitivity experiment shows a pattern very similar to simulation HIGH, characterized by a large FW storage in the Beaufort Gyre and a decrease of FW content from the Canadian Basin towards the Eurasian Basin as expected from observations (Fig. 10d). In the





**Figure 9.** (a) The time series of annual mean total Arctic liquid freshwater (FW) content. The liquid FW content in Eurasian Basin, Canadian Basin and shelf regions are shown in (b)(c)(d), respectively. The FW content is calculated using a reference salinity of 34.8.



simulation LOW, because the CAA straits are poorly resolved with coarse resolution and the CAA outflow is restricted (see more details in the section of mean state), more FW takes the release route through Fram Strait, thus increasing the FW content in the western Eurasian Basin. Therefore, it is mainly resolving the narrow straits in simulation HIGH that leads to the FW content difference from simulation LOW, rather than the high resolution inside the Arctic Ocean.

## 4.3  Mean state of liquid freshwater

As a consequence of salinity drift during the model spin-up, the basin mean salinity shows biases in the halocline in both Arctic basins (Fig. 5b). The biases are largest at the mid-depth of the halocline, as the salinity is restored to the climatology at the ocean surface, and below the halocline the salinity is determined by that of the AW. As the Eurasian Basin bias in simulation LOW is larger than in simulation HIGH, the overestimation of Arctic FW content is more significant in LOW ( 26% compared
to 18%, Table 1). As mentioned above, the spatial distribution of liquid FW content is better reproduced in HIGH than in LOW (Fig. 10a-d), albeit with overestimation in both simulations, because the high resolution more faithfully represents the narrow channels in CAA. The variety of FW content distributions simulated in different ocean models shown by *Wang et al.* (2016b) presumably can be partly attributed to different model representations of the CAA region.

The spatial pattern of FW content is manifested in the simulated sea surface height (SSH, see Fig. 11), since the steric height
is dominated by the halosteric component in the Arctic Ocean (e.g., *Griffies et al.*, 2014). In simulation HIGH the SSH field shows a better represented Beaufort Gyre. The CAA resolution not only impacts the FW content pattern, but also the circulation and export pathways of water masses. For example, as illustrated by passive tracers (Fig. 11), in simulation LOW the Pacific Water penetrates more into the Canadian Basin, and has a higher concentration at Fram Strait than in HIGH. And the better resolved CAA channels in HIGH allow more Atlantic Water from BSO to be released through the CAA.

To access the simulated mean state of FW transport through main Arctic gateways, we compare the model results for the period of 1980-2000 with the synthesized values by *Serreze et al.* (2006, see Table 1). Observations suggested that more FW is released through the CAA than through the Fram Strait. This is reproduced in simulation HIGH, while the FW transports through the two export gateways are nearly the same in LOW. Although the simulated CAA FW export in both simulations is lower than the synthesized value, the CAA FW export in HIGH is significantly higher than in LOW, and still within the
observational uncertainty range. At Fram Strait both the ocean volume and FW transports in LOW are higher than in HIGH, as expected from the impact of resolution in the CAA discussed above. Although using higher resolution reduces the Fram Strait FW export, the mean value is still close to the lower bound of the observational range. At Bering Strait the FW import is underestimated in the two simulations, with simulation HIGH obtaining a slightly higher value, very close to the lower bound of the observational range. As the Bering Strait ocean volume transports in the two simulations are within the range suggested by observations, the underestimation of FW transports is due to biases in Pacific Water salinity, which could be still in a phase
of large scale spin-up within the whole model integration period.



**Table 1.** Arctic Ocean liquid and solid freshwater (FW) budget relative to a reference salinity of 34.8, and the net ocean volume transport through Arctic gateways. The FW budget terms are shown for the periods 1980-2000 and 2000-2009 separately. The correlation coefficients for fluxes obtained from the two simulations (LOW and HIGH) are shown for the period 1980-2009 in the last column, and all correlations are significant at the 95% level. Liquid FW contents for the 2000-2009 period are shown with the **changes** relative to the 1980-2000 period. FW fluxes are shown in $\mathrm{km^3/year}$, FW contents are in $10^4\,\mathrm{km^3}$, and ocean volume transports are in Sv. Positive fluxes indicate sources for the Arctic Ocean.

| | **1980-2000** | | | **after 2000** | | | Model |
| | Observation | LOW | HIGH | Observation | LOW | HIGH | correlation |
|---|---|---|---|---|---|---|---|
| **Liquid freshwater** | | | | | | | |
| Fram Strait | $-2660 \pm 528^a$ | -2306 | -2115 | $-2800 \pm 420^b$ | -1979 | -1861 | 0.78 |
| Davis Strait | $-3200 \pm 320^a$ | -2263 | -2887 | $-2900 \pm 190^b$ | -2199 | -2722 | 0.75 |
| Bering Strait | $2400? \pm 300^a$ | 2029 | 2170 | $2500 \pm 100^b$ | 1932 | 2079 | 0.98 |
| BSO | $-90 \pm 94^a$ | -591 | -441 | $-90 \pm 90^b$ | -779 | -664 | 0.90 |
| Arctic FW content | $6.92\ ^c$ | 8.69 | 8.19 | $(+0.45)^{b,d}$ | $(+0.17)$ | $(+0.17)$ | |
| **Solid freshwater** | | | | | | | |
| Fram Strait | $-2300 \pm 340^a$ | -2369 | -2488 | $-1900 \pm 280^b$ | -2065 | -2154 | 0.95 |
| Davis Strait | $-160 \pm ?^a$ | -416 | -427 | $-320 \pm 45^b$ | -320 | -342 | 0.98 |
| NH FW content | $1.8^e$ | 2.28 | 2.21 | $1.44^e$ | 1.84 | 1.81 | |
| | **1980-2009** | | | | | | Model |
| | Observation | LOW | HIGH | | | | correlation |
| **Ocean volume flux** | | | | | | | |
| Fram Strait | $-2 \pm 2.7^f$ | -2.18 | -1.84 | | | | 0.88 |
| Davis Strait | $-3.2 \pm 1.2$ to | -1.03 | -1.69 | | | | 0.90 |
| | $-1.6 \pm 0.2^g$ | | | | | | |
| Bering Strait | $0.8 \pm 0.2^{h,i}$ | 0.87 | 0.95 | | | | 0.99 |
| BSO | $2.0$ to $2.3^{j,k,l}$ | 2.36 | 2.52 | | | | 0.93 |

[a] *Serreze et al.* (2006), [b] *Haine et al.* (2015), [c] computed from PHC3 (*Steele et al.*, 2001), [d] *Polyakov et al.* (2013b), [e] based on the PIOMAS Arctic sea ice volume reanalysis (*Schweiger et al.*, 2011) by assuming sea ice density of $910\,\mathrm{kg/m^3}$ and salinity of $4\,\mathrm{psu}$, [f] *Schauer et al.* (2008), [g] *Curry et al.* (2014), [h] *Roach et al.* (1995), [i] *Woodgate and Aagaard* (2005), [j] *Smedsrud et al.* (2010), [k] *Skagseth et al.* (2008), [l] *Smedsrud et al.* (2013).



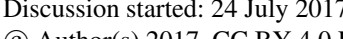

**Figure 10.** Mean liquid freshwater (FW) content (in m) for the period 1993–2002 for (a) LOW, (b) HIGH, (c) CAA HIGH, and (d) observation of *Rabe et al.* (2011). Difference in liquid FW content between the periods 2003–2007 and 1993–2002 for (e) LOW, (f) HIGH, (g) CAA HIGH, and (h) observation. Linear trend of FW content (m/decade) for the period 1996–2009 for (i) LOW, (j) HIGH and (k) CAA HIGH. The FW content is calculated using a reference salinity of 34.8. The sensitivity experiment CAA HIGH is introduced to isolate the impact of resolution in the Canadian Arctic Archipelago (CAA) from that in the Arctic interior. It has resolution similar to LOW outside CAA, but the same resolution as in HIGH inside CAA.





## (a) 24 km

## (b) 4.5 km

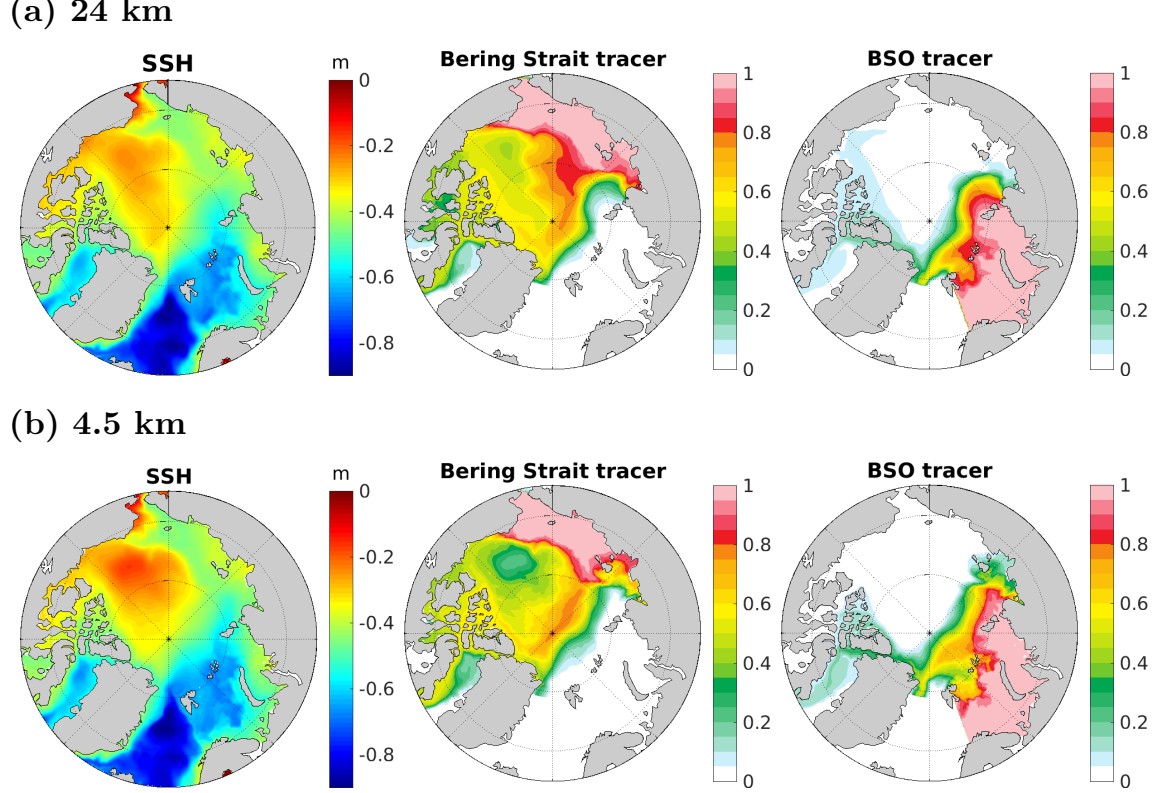

**Figure 11.** (a) Mean sea surface height (left), Bering Strait passive tracer (middle), and Barents Sea Opening (BSO) passive tracer (right) in simulation LOW for the period of 1993–2002. (b) The same as (a) but in simulation HIGH. Note that the passive tracers were set to zero south of Fram and Davis Straits. The passive tracers are averaged over the upper 100 m.

### 4.4 Variability of liquid freshwater

The simulated liquid FW contents do not show significant interannual variability, but rather large decadal changes (Fig. 12a). In both simulations the FW content decreases from the beginning of 1980s until the mid-1990s, and then increases afterwards. The descending trend of observed FW content (*Polyakov et al.*, 2013b) before the mid-1990s is much lower (Fig. 12a). Most of

5  the models used in the CORE-II model intercomparison obtained a significant descending trend before the mid-1990s (figure 8 of *Wang et al.*, 2016b), as in the two simulations presented here. Compared to the period of 1980-2000, the mean Arctic FW content averaged over the 2000s has increased by about $4500\,\text{km}^3$ based on observations (*Polyakov et al.*, 2013b; *Haine et al.*, 2015), while the increase is only about $1700\,\text{km}^3$ in our two simulations (Table 1).

     The linear trend in the FW content for the period 1996–2009 based on the data set of *Polyakov et al.* (2013b) shown in Fig.

10  12a is $844\,\text{km}^3/\text{yr}$. The upward trends in the two simulations are lower, having half of this value in LOW ($423\,\text{km}^3/\text{yr}$) and 60% of it in HIGH ($521\,\text{km}^3/\text{yr}$). On average the 13 CORE-II models analyzed in *Wang et al.* (2016b) underestimated the



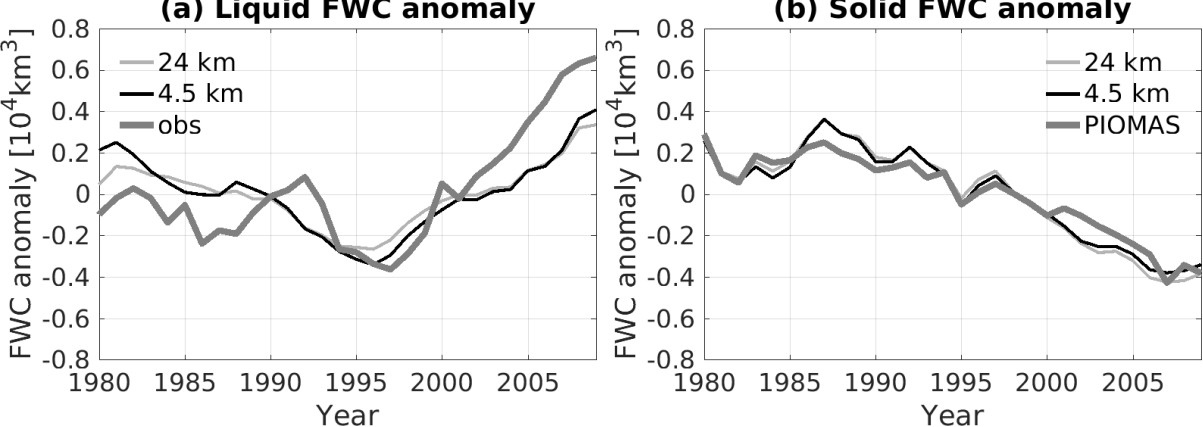

**Figure 12.** Anomalies of (a) liquid freshwater (FW) content and (b) solid FW content of the Arctic Ocean. The FW content is calculated using a reference salinity of 34.8. The liquid FW content observation is provided by *Polyakov et al.* (2013a), and the solid FW content is compared to the data derived from PIOMAS reanalysis (*Schweiger et al.*, 2011).

observed upward trend also by half. Although the total Arctic liquid FW content increases nearly linearly after the mid-1990s, the situation is quite different in the individual Arctic basins. In both simulations, during the last 5 years of the integration, the upward trend strengthens in the Canadian Basin, while the trend almost stops in the Eurasian Basin, and a descending trend is non-negligible over the continental shelves (Fig. 9). The model result is consistent to the observed scenario of changes in

FW distribution in the two Arctic basins described by *Morison et al.* (2012). They explained that the changes were due to a cyclonic shift in the ocean pathway of Eurasian runoff associated with an increased Arctic Oscillation index.

    We are also interested in the model representation of temporal variation of FW content spatial distribution. In Fig. 10e-h the difference in FW content between the periods of 2003-2007 and 1993-2002 is shown. The observation indicates that the most significant increase in FW content between the two periods is along the Chukchi Sea continental slope and on the periphery of

the Beaufort Gyre. At the latter location the simulations did not obtain a similar pattern of positive changes. The FW content increases on both side of the Lomonosov Ridge in the observation. Simulation HIGH consistently obtains positive changes in the Eurasian Basin, but with a larger magnitude. It has negative values north of Greenland, which is not present in the observation.

    The spatial pattern of positive changes in FW content in HIGH is very similar to that obtained in a model with about 12 km

resolution in the Arctic Ocean shown by *Wang et al.* (2016b). In their study most other models show a quite different pattern because of too coarse model resolution used (about 1 degree resolution). Besides the difference in FW content between the two periods, we also calculated the linear trend of 2D FW content from 1996 to 2009 (Fig. 10i-k). The two methods of diagnosing the temporal variation of 2D FW content provide similar conclusions on the impact of model resolution (compare Fig. 10e-g with Fig. 10i-k). As mentioned above, the resolution inside the CAA plays an important role in representing the mean state of

the Arctic Ocean FW content. Here the additional sensitivity experiment, where high resolution is only applied in the CAA





**Figure 13.** Anomalies of annual mean freshwater (FW) transport through main Arctic gateways. Liquid FW transport through (a) Fram Strait, (b) Davis Strait, (c) Bering Strait and solid FW transport through (d) Fram Strait. The dotted lines show the multi-model means (MMM) obtained from 13 CORE-II models (*Wang et al.*, 2016a, b).

channels, helps to illustrate that the high resolution inside the Arctic Ocean does have some impact on the representation of 2D FW content variation, for example, in the Beaufort Gyre and the central Eurasian Basin.

The interannual variability of FW transport through the Arctic gateways shows large similarity between the two simulations (Fig. 13a-c). The correlation coefficients between the FW transports from the two simulations are similar at Davis and Fram Straits (0.75 and 0.78, respectively, Table 1). The correlation is lower than the correlation for ocean volume transports, indicating that the simulated interannual variability of salt transport changes between the two simulations and leads to reduced





inter-simulation correlation for FW transports. The current model results are largely similar to the multi-model mean result analyzed by *Wang et al.* (2016b, also plotted in Fig. 13). The most significant difference is in the Fram Strait FW transport. For example, the changes of FW transport from the mid-1990s to the beginning of 2000s is more pronounced in our two simulations (Fig. 13a). The variability of FW transport at Fram Strait has been found to be the least consistently simulated
among both AOMIP and CORE-II models (*Jahn et al.*, 2012; *Wang et al.*, 2016b). At Bering Strait the variability is nearly not distinguishable between the two simulations and the multi-model mean obtained in the past model study (Fig. 13c).

On decadal time scales, the observed FW export through Davis Strait in the 2000s is about $10\%$ lower than the climatology of 1980-2000 (*Haine et al.*, 2015). Both simulations reproduce the reduction in the Davis Strait FW export, but the magnitude of reduction is less significant than the observed (Table 1). In simulation HIGH the reduction (about $5\%$) is larger than in
LOW. At Fram Strait the FW export is suggested to be slightly higher in the 2000s than in the period of 1980-2000 (*Haine et al.*, 2015), while the two simulations similarly show an opposite result, obtaining a reduction of $\sim 300\,\mathrm{km}^3/\mathrm{yr}$ in 2000s. The Bering Strait FW transport remains nearly at the same level after 2000s, which is reproduced by the simulations. Note that the uncertainty in observations is large due to the sparseness of measurements, and both the observed and simulated changes in FW transports through the Arctic gateways between the two periods are smaller than the magnitude of respective observational
uncertainty.

## 4.5 Sea ice and solid freshwater

The sea ice volume (and corresponding solid FW content) in the two simulations is nearly the same (Table 1), because both the sea ice thickness and concentration are not significantly influenced by the model resolution (Fig. 14). At 4.5 km the sea ice model starts to capture some small scale features (sea ice leads) with reasonable spatial and temporal variability (*Wang*
*et al.*, 2016c). However, the mean sea ice thickness and concentration is not impacted by whether those small scale features are represented or not in the model. Note that much higher model resolution is required in order to simulate sea ice leads with realistic width.

The summer sea ice area along the sea ice edge on the Eurasian side is slightly overestimated in both simulations, and the simulated sea ice thickness is about half a meter thicker than the satellite observation in the last few model years (Fig. 14).
Because of lacking sufficient long-term sea ice thickness observations, we compare our simulated solid FW content with the estimate from the PIOMAS Arctic sea ice volume reanalysis (*Schweiger et al.*, 2011). The simulated mean solid FW content in the period of 1980-2000 is about $20\%$ higher than the PIOMAS estimate (Table 1).

The time series of annual mean solid FW content show that the two simulations obtain a descending trend very similar to that from the PIOMAS estimate (Fig. 12b). Compared to the mean value before 2000, the solid FW content decreases by about
$4000\,\mathrm{km}^3$ averaged over the 2000s in the simulations, similar to the PIOMAS result (Table 1). Because our simulated sea ice thickness is underestimated before 2000 compared to the submarine observations (as shown in figure 13 of *Wang et al.*, 2016a) and overestimated in later years compared to satellite observations (Fig. 14a-c), the descending trend of solid FW content over the last 3 decades in our simulation and in the PIOMAS estimate as well might be lower than reality.





**Figure 14.** Spring sea ice thickness averaged from 2004 to 2007 for (a) simulation LOW, (b) HIGH, and (c) the ICESat observation (*Kwok et al.*, 2009). September sea ice concentration averaged from 1979 to 2009 for (d) simulation LOW, (e) HIGH, and (f) the NSIDC observation (*Fetterer et al.*, 2016). In (d),(e) the black curves show the 15% contour lines of the observed sea ice concentration, while the red curves show those of simulations.

Arctic sea ice is mainly exported through Fram Strait. The two simulations produced very similar solid FW transports through Fram Strait, well representing the observed value (Table 1). Although the sea ice area export through Fram Strait has been increasing in recent decades due to increasing sea ice drift (e.g., *Smedsrud et al.*, 2017), sea ice volume and thus solid FW export has been decreasing due to the thinning of Arctic sea ice. Compared to the estimate of the 1980-2000 period, the solid FW export flux decreased by $400 \, \mathrm{km}^3/\mathrm{yr}$ in the 2000s (*Haine et al.*, 2015). The two simulations similarly produce a decrease in Fram Strait solid FW export of about $300 \, \mathrm{km}^3/\mathrm{yr}$ between the two periods (Table 1). On interannual time scales the two simulated solid FW transports are well correlated (Table 1 and Fig. 13d). As shown in *Wang et al.* (2016a, b), ocean climate models can more consistently simulate the interannual variability of solid FW transports through Arctic gateways than the liquid FW transports.





## 5 Discussion

### 5.1 Atlantic Water

### (a) Heat content and water mass sources

In simulation LOW the warm AW occupies a much thicker layer than in simulation HIGH (Fig. 5). The thickening and deep-
ening of the AW layer in simulation LOW can be explained by spurious mixing associated with tracer advection schemes, as
suggested in past studies (*Holloway et al.*, 2007). If there was no additional net heat source in simulation LOW, the magnitude
of the temperature in the AW layer would be reduced because the heat would be distributed in a much larger ocean volume. Our
model results show that the magnitude of the AW temperature is not lower in LOW. At the end of the simulations the Arctic
heat content is higher than the climatology in both simulations, but it is about $4 \times 10^{21}$ J higher in simulation LOW than in
HIGH. This difference in heat content requires an additional heat flux of 2 TW over 60 years. Due to inaccuracy in diagnosing
heat budget terms (e.g., caused by interpolation) and ignoring heat diffusion in model output, the mismatch between the ocean
heat content changing rate and Arctic net heat flux can have the same order of magnitude as this value. Therefore it is hard to
carry out analysis of closed heat budget in this and previous modeling studies (e.g., *Lique and Steele*, 2013).

The temperature and heat content in the AW layer is influenced by both the warm Fram Strait and the cold BSO AW branches,
the latter of which joins the former mainly through the St. Anna Trough (*Schauer et al.*, 2002). Using passive tracers we can
obtain the spatial distribution of the two water sources (Fig. 15). The locations of the maxima of the Fram Strait passive tracer
coincide with the maxima of temperature in both basins (cf. Fig. 15a and Fig. 5a). The maxima of the Fram Strait passive tracer
are located deeper in simulation LOW than in HIGH, consistent to the deepening of the AW layer shown by its temperature
maxima. Below about 350 m depth the concentration of Fram Strait passive tracer in LOW is higher in both Arctic basins than
in HIGH (Fig. 15a,c). In HIGH the Fram Strait passive tracer has weaker penetration into the Canadian Basin, and a stronger
cyclonic circulation inside the Eurasian Basin. At the end of 2000 the Fram Strait passive tracer averaged over the whole Arctic
volume in LOW is about $14\%$ higher than in HIGH. As the volume import of the Fram Strait AW in HIGH is larger (calculated
at 79°N in the Fram Strait), a lower passive tracer storage implies that the export of Fram Strait branch AW is stronger in
HIGH, either via direct recirculation north of Fram Strait or after cyclonic circulation in the Eurasian Basin,

The BSO passive tracer indicates that cold AW (lower than 0°C) from the BSO has a lower concentration in the Eurasian
Basin in simulation LOW than in HIGH, and the situation is opposite in the Canadian Basin (Fig. 15b,d). AW from both
branches have replenished the Canadian Basin more intensively in simulation LOW. BSO AW has the effect to reduce the
temperature of the AW layer, so the higher temperature and heat content in the Canadian Basin in LOW should be attributed to
the larger amount of warm Fram Strait AW. Because the temperature of the BSO branch is similar after the atmospheric cooling
over the continental shelves, the slightly lower volume transport through BSO in LOW (Table 1) has a positive contribution to
the overall AW layer heat content.



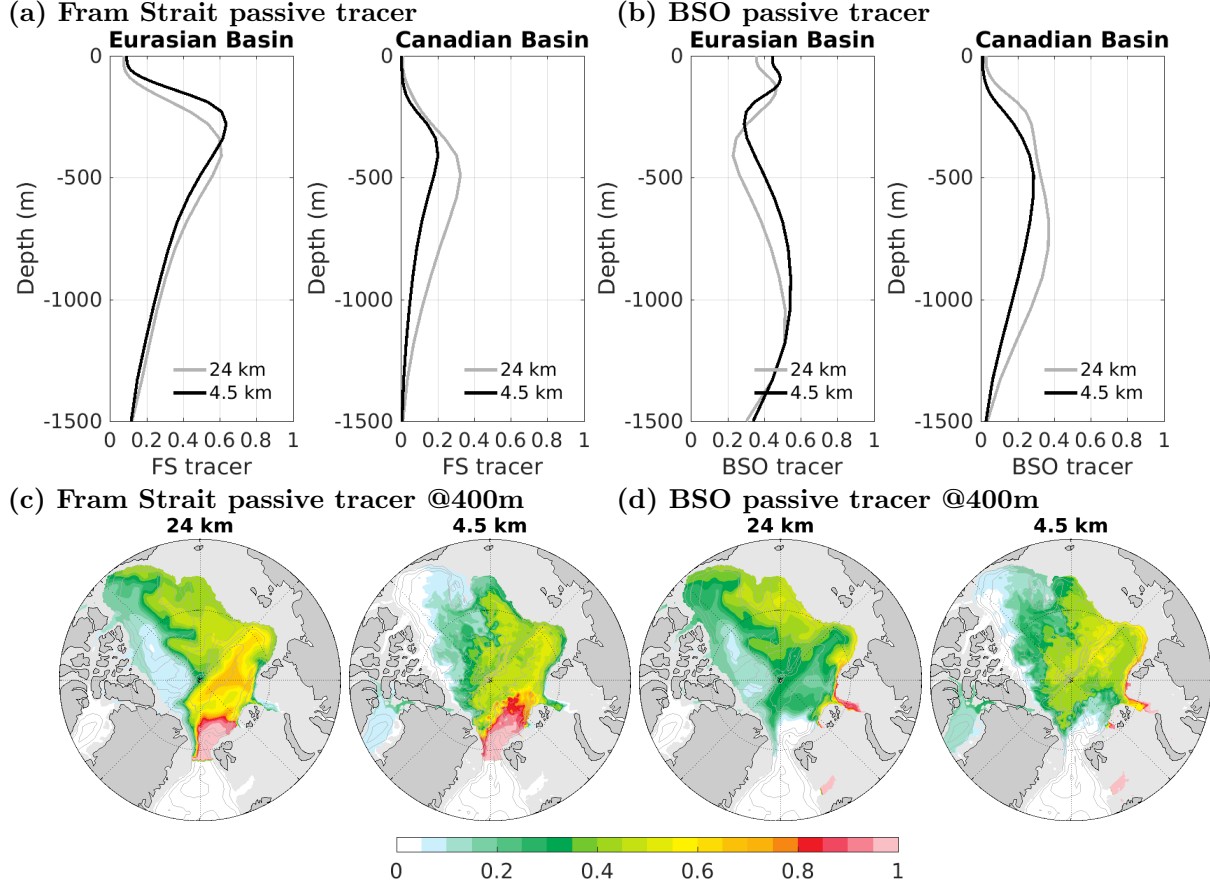

**Figure 15.** (a) Mean Fram Strait passive tracer concentration in the two Arctic basins averaged over the year 2000. (b) The same as (a) but for the Barents Sea Opening (BSO) passive tracer. (c) The Fram Strait passive tracer at 400 m depth averaged over the year 2000. (d) The same as (c) but for the BSO passive tracer.

**(b)    Future work related to simulating AW**

Fram Strait is the main pathway of oceanic heat flux from the North Atlantic into the Arctic basins. It is very challenging for numerical models to simulate the complex AW circulation in Fram Strait. In the few degree latitude band the AW loses heat due to surface cooling and starts to subduct under cold Polar Water, and a fraction of AW recirculates to the west and then

5    southwards in different paths (*Quadfasel et al.*, 1987; *Gascard et al.*, 1988; *Saloranta and Haugan*, 2001; *Marnela et al.*, 2013; *de Steur et al.*, 2014). Strong variability associated with mesoscale eddies was observed in the Fram Strait (*von Appen et al.*, 2016), which may play an important role in setting the AW recirculation (*Hattermann et al.*, 2016). The first baroclinic Rossby radius in Fram Strait is very small (about 2 km in winter), thus our high resolution (4.5 km grid size) simulation cannot resolve mesoscale eddies. At this resolution the simulated warm AW is confined to the strong boundary current and does not reach





the central Fram Strait, presenting a cold bias in the center of the strait (*Wekerle et al.*, 2017). As in other high resolution, but not eddy-resolving models (e.g., *Fieg et al.*, 2010), our simulated AW temperature in the boundary current is too high at Fram Strait and north of Svalbard (Fig. 4a). The deficiency indicates a clear requirement for eddy resolving resolution in the Fram Strait region in order to faithfully simulate the amount and property of AW that enters the Arctic basins through the Fram Strait.

In ocean climate simulations, however, it is hardly possible to afford 1 km model resolution in the near future. Accordingly, efforts on parameterizations are required to improve the simulation of AW circulation in Fram Strait.

The AW is located at intermediate depths in the Arctic Ocean and is separated from surface water and sea ice by a strong halocline. However, recent pan-Arctic microstructure measurements of turbulent kinetic energy dissipation reveal that tides can significantly enhance vertical mixing and bring up substantial heat in some areas (*Rippeth et al.*, 2015), implying an impact

of AW heat on Arctic sea ice. It was shown that tides can explain a non-negligible part of the sea ice volume reduction in numerical simulations (*Luneva et al.*, 2015). Tides are not simulated in our model, so their potential impact on sea ice and AW characteristics is not explicitly considered. If tides were present in the simulations, and indeed have significant impact on heat uptaken, the influence of AW on sea ice would be different in the two simulations, because the temperature and depth of the AW layer are different between them. Dedicated studies are required to investigate such effect.

After the AW warming in the Arctic basins in the 1990s, unprecedented warming has been observed in the 2000s (*Polyakov et al.*, 2013b). However, no warming as strong as observed was obtained in the latter period in the two model simulations (Fig. 3). The AW transport calculated in the northern Fram Strait was found to decrease in the 2000s in the simulations. As the warming in 1990s is reasonably represented in the model, the discrepancy between the observed and simulated temperature variation in recent years could be attributed to model deficiency in representing ocean processes under the condition of sea ice

decline, or to the quality of the atmospheric forcing data used. Furthermore, the AW layer temperature in the Arctic interior is not only determined by the amount of warm AW through Fram Strait and cold AW from BSO, but also the circulation details of the two branches inside the basins. Research on these subjects is required in future work.

### 5.2 Freshwater

#### (a) Freshwater content drift and sea surface salinity restoring

In both simulations the Arctic liquid FW content increases rapidly during the first 20-30 years, the same as in other ocean climate models participating in the CORE-II intercomparison project analyzed by *Wang et al.* (2016b). They showed that the source of excessive FW is sea surface salinity (SSS) restoring. Here we repeated the low resolution simulation with SSS restoring switched off. In this simulation, the salinity in the Canadian Basin obtains a positive bias instead of a negative one, most pronounced at the surface (Fig. 5c). In the Eurasian Basin the salinity bias is still negative, but becomes smaller. The

spin-up in this sensitivity run also takes about 20 to 30 years (Fig. 9). The FW content in Canadian Basin decreases in the spin-up phase, with a magnitude similar to that of FW content increase in the two simulations with SSS restoring (Fig. 9c). In the Eurasian Basin the FW content remains lower than in LOW by nearly a constant offset after 30 model years (Fig. 9b). The total Arctic FW content does not have a significant model drift (Fig. 9a), because the opposite drifts in the two basins largely





cancel each other. In the last 30 model years, the variability of FW content in both basins in the sensitivity simulation is similar to that in simulation LOW and HIGH.

In the sensitivity simulation without SSS restoring, the salinity has a positive bias at the surface and negative bias in the lower halocline in the Canadian Basin (Fig. 5c). This implies that too much vertical mixing has taken place, which could be linked to

the fact that brine rejection induced convection on very small spatial scales is neither resolved nor properly parameterized in the model: If salt rejected during ice formation is added to the ocean surface, the static instability on the model grid may initialize strong vertical mixing and weaken the vertical salinity gradient, resulting in negative salinity anomaly in the halocline and positive salinity anomaly near the ocean surface. The ocean temperature profile in this depth range is also smoothed out. This issue was discussed by, for example, *Duffy et al.* (1999) and *Nguyen et al.* (2009), who proposed to distribute rejected salt in

the ocean column with some vertical distribution function, thus preventing static instability. By doing so they got significantly improved salinity profiles. We have implemented this parameterization for brine rejection in the model and are able to achieve improvement on the salinity representation in the Canadian Basin. However, it is not easy to define one particular salt vertical distribution function that can satisfy different Arctic basins and the Southern Ocean at the same time. Some research is required before we can suggest a default scheme for brine rejection in our global model simulations. The background vertical diffusivity

was suggested to be one of the key parameters controlling the simulated Arctic Ocean hydrography and circulation, especially in the Canadian Basin (*Zhang and Steele*, 2007; *Nguyen et al.*, 2009). In our next model tuning phase, FESOM sensitivity to such model parameters should be carefully examined.

The salinity bias and overestimated FW content in the Eurasian Basin is very possibly caused by inaccurate representation of the pathways of upper ocean circulation (Fig. 8e,f). The Transpolar Drift carrying fresh Pacific Water and river water is

located too much to the Eurasian side of the Lomonosov Ridge, and the anticyclonic surface circulation in the Canadian Basin occupies a too large spatial range compared to the observation (Fig. 10a-d). The low resolution inside CAA in simulation LOW causes more FW to release through Fram Strait, which further increases the FW content in the Eurasian Basin. In the sensitivity simulation without SSS restoring, the SSS is still well represented in the Eurasian Basin. The Eurasian Basin salinity bias in the halocline becomes smaller in this sensitivity simulation, because the FW content is lower and the anticyclonic circulation

shrinks in the Canadian Basin, with less FW penetrating into the Eurasian Basin. The upper ocean circulations are mainly driven by surface wind stress, so it is required to investigate the wind forcing fields and the impact of sea ice on the ocean surface stress in order to better understand the Eurasian Basin salinity drift.

**(b)  Basinwise and Beaufort Gyre freshwater content variability**

In this work we have assessed the total Arctic FW content and its distribution between the Eurasian and Canadian Basins. It was

found that the increase of FW storage in the Canadian Basin in recent years behaves nearly identically in different simulations (Fig. 9c). On the contrary, the trend of FW content in the Beaufort Gyre region indicates difference among the simulations (Fig. 10i-k). Recent research indicates that mesoscale eddy fluxes counteract Ekman pumping, thus playing a crucial role in Beaufort Gyre FW content variability (e.g., *Manucharyan et al.*, 2016; *Yang et al.*, 2016). In model simulations eddy parameterization (applied on coarse meshes) and the effect of implicit numerical mixing will certainly influence the dynamical balance and the





Beaufort Gyre FW content. Further effort is required to investigate the model representation of Beaufort Gyre FW content and more importantly its relationship to Arctic FW release to the North Atlantic.

## 5.3 Unstructured-mesh modeling

The variable-resolution functionality provided by unstructured-mesh models offers new possibility in ocean modeling. One can increase model resolution locally where research interest is located, without necessity of using traditional nesting. On the mesh the resolution can vary in space conveniently according to given functions chosen for particular applications. Many ocean process studies have been carried out taking use of FESOM in global and regional simulations, for example, with focus on overflows (*Wang et al.*, 2012), ice shelf cavities (*Timmermann et al.*, 2012), deep water formation (*Scholz et al.*, 2013), polynyas (*Haid and Timmermann*, 2013), and Arctic Ocean dynamics (*Wekerle et al.*, 2013). In global ocean climate simulations, the value of unstructured meshes can be more outstanding. One can design meshes with resolution varying continuously in space according to the strength of ocean variability, for example, by considering observed sea surface height variability (*Sein et al.*, 2016) and/or Rossby radius, to permit or resolve mesoscale eddies in mid to low latitudes. It would be interesting to use this kind of global meshes together with specific mesh refinement in the Arctic Ocean for the purpose of Arctic Ocean studies, as the lower latitude ocean will be better resolved with acceptable increase of computational cost and provide more faithful oceanic linkage with the Arctic Ocean. Developing such a model configuration is aligned with our strategic plan for Arctic Ocean modeling using FESOM and the coupled climate model. It will facilitate us to study and predict not only Arctic changes, but also large scale linkage between high and lower latitudes. Towards this goal, we need to understand, for example, the impact of regional resolution in the Arctic region, using economy configurations as reported in this paper.

An adequate representation of the CAA throughflow is found to be very important. With an unstructured-mesh model like FESOM, one can locally increase model resolution to accurately resolve the narrow channels and faithfully simulate the FW export (*Wekerle et al.*, 2013). However, if the finest grid size is used in narrow straits in FESOM, the model time step and the overall model throughput could be constrained by this grid size. In ocean climate simulations, therefore, it is not preferable to have model resolution in narrow straits higher than the highest resolution used in large ocean basins. In simulation HIGH, the same high resolution is used in the Arctic Ocean and the CAA channels, so the model time step is set by the model stability outside the CAA region because largest velocities occur outside the CAA. In simulations like LOW, practically we do not try to further increase resolution in the CAA as done in one of the sensitivity experiments used in this work. Instead, we often modify the geometry of the CAA channels to allow adequate CAA throughflow. Such model adjustment, however, is not trivial as shown by the large model spread in CAA FW transports among the ocean climate models analyzed in *Wang et al.* (2016b). When developing global climate models, the modeling groups certainly need more efforts to better adjust the CAA representation. Besides, maintaining high resolution measurements of ocean transports is of great importance for model development too.

In the structured-mesh model community global and near-global ocean models with mesoscale-eddy resolving resolutions have been developed in many groups (e.g., *Chassignet et al.*, 2009; *Storch et al.*, 2012; *Oke et al.*, 2013; *Metzger et al.*, 2014; *Dupont et al.*, 2015; *Iovino et al.*, 2016), and coupled climate models with eddy resolving ocean have also been used



in practice (e.g., *Griffies et al.*, 2015). Most of the models analyzed in past CORE-II model intercomparison studies have relatively coarse resolution. For developing our unstructured-mesh model system with regional focus, it would be helpful to communicate experience with the large structured-mesh model community, for example, in future high resolution ocean climate model intercomparison projects.

5 ## 6   Summary

A faithful model representation of the ocean circulation, water mass property and sea ice state in the Arctic Ocean is still challenging, not only for its mean state, but also for the variability of some of the key diagnostics, in state-of-the-art ocean-sea ice models (e.g., *Jahn et al.*, 2012; *Wang et al.*, 2016a, b; *Ilicak et al.*, 2016). With the development of computing resources and model technology, high resolution Arctic Ocean modeling starts to become affordable even in ocean climate simulations. In 10 this work we explored the impact of high horizontal resolution on the circulation of the Atlantic Water (AW) in the intermediate layer and freshwater (FW) in the upper layer of the Arctic Ocean. In particular, the mean state and variability of the AW layer and the Arctic FW budget are assessed, for which previous model intercomparison studies have provided basic knowledge on common model issues.

The simulations of the unstructured-mesh ocean-sea ice model FESOM (*Wang et al.*, 2014) with two global meshes differing 15 in resolution in the Arctic Ocean are evaluated. The coarse resolution mesh has been used in previous CORE-II model inter-comparison studies (e.g., *Griffies et al.*, 2014; *Danabasoglu et al.*, 2014). Its resolution in the Arctic Ocean is 24 km. On the high resolution mesh the Arctic resolution is increased to 4.5 km. In most parts of the Arctic Ocean this resolution is not eddy resolving. As our intention is to provide information for developing model configurations that can be used for ocean climate simulations, a reasonably high model throughput is a prerequisite. With 4.5 km resolution in the Arctic Ocean we can run 20 FESOM for about 7 model years per day. Using further higher resolution, though preferable for the Arctic region due to very small Rossby radius, would prevent us and groups working on other ocean climate models from carrying out long simulations at the current stage. For ocean process studies, we certainly can use the variable resolution functionality of FESOM to even better resolve local dynamics (for example, using 1 km horizontal grid size locally to resolve mesoscale eddies in the Fram Strait, Wekerle et al., in preparation). This aspect of Arctic Ocean modeling is beyond the scope of this paper. As we kept the 25 same model resolution outside the Arctic region, we are able to attribute the difference in the two simulations to the Arctic Ocean resolution. Note that we did not try to tune the two model setups separately in this paper. We used a model configuration (schemes and parameters for ocean and sea ice) similar to what has been used in the CORE-II studies (e.g., *Danabasoglu et al.*, 2014), except that eddy diffusivity is scaled by the resolution.

At 24 km resolution the simulated AW layer is unrealistically deep and thick, which currently is a common issue in coarse 30 resolution models (*Ilicak et al.*, 2016). Such a model bias was found to be caused by numerical mixing in past AOMIP studies (*Holloway et al.*, 2007). When using 4.5 km resolution, the AW in both Arctic basins is located at the observed depth with a very reasonable thickness. The tracer advection scheme (a second order FCT scheme) used in our simulations is the one suggested for large scale applications in FESOM, because it enforces monotonicity and has decent computational cost. Idealized 2D




test cases clearly indicate that numerical smoothing associated with this scheme can be significantly reduced with increasing resolution (*Wang*, 2007), which can explain the obtained improvement of the AW layer in the high resolution simulation. As we kept the vertical resolution the same in the two simulations, which needs separate investigation, the reduction in numerical mixing is only due to the change in horizontal resolution.

5    With higher resolution the cyclonic AW boundary current becomes narrower and more energetic. Moreover, the topographic steering on the current is stronger, causing more AW to recirculate along the Lomonosov Ridge in the Eurasian Basin. The resulting constrained penetration of AW into the Canadian Basin in the high resolution simulation helps to eliminate the intensive warming and deepening trend of the AW in the Canadian Basin present in the coarse resolution setup. More AW recirculates in the Eurasian Basin and leaves the Arctic Ocean, which can partly explain that the increase in Arctic heat content is much lower in the high resolution simulation. The strength of interannual and seasonal variability of AW temperature, especially in the boundary current along the continental slope and Lomonosov Ridge, becomes significantly higher with increasing resolution.

The impact of horizontal resolution on ocean surface circulation and FW cycle is limited to the spatial pattern of liquid FW content and pathways of different water masses. It mainly stems from the difference in the representation of the Canadian Arctic Archipelago (CAA) channels, not the resolution in the Arctic basins. The CAA channels are often treated very differently in different ocean climate models, for example, for the number of CAA channels and number of active grid points across the channels, as shown in the model intercomparison study by *Wang et al.* (2016b). They found that the spread in simulated CAA and Fram Strait FW transports is considerably large. Therefore, inspecting and tuning CAA representation is one of the important tasks in future development of ocean climate models.

The main state and variability of total and basinwise liquid FW content, the variability of liquid FW transports through Arctic gateways and the characteristics of Arctic sea ice volume and export do not change significantly with increasing resolution. The recent upward trend of FW content in the Beaufort Gyre shows some sensitivity to the resolution inside the Arctic basin. How well mesoscale eddies are resolved in the Canadian Basin in the high resolution simulation, and how realistic the effect of eddies is parameterized in the low resolution simulation, both need to be assessed in the context of the interplay with Ekman pumping in future studies. Here it is important to note that the variability of both the Arctic FW storage and release to North Atlantic is insensitive to the model resolution applied in our simulations.

Besides identifying the impact of horizontal resolution on the Arctic Ocean main circulation, we also discussed scientific questions and model issues that need to be explored in future work, and some of the illustrated model issues are common in many other ocean-sea ice models. Overall, increasing model resolution does considerably improve the performance of the Arctic Ocean simulation, while further efforts are necessary to solve remaining issues that are not linked to applied model resolution, and to develop/improve parameterizations that are still required even with best resolution affordable now.

*Code and data availability.* FESOM v1.4 can be downloaded from https://swrepo1.awi.de/projects/fesom after registration. For the sake of the journal requirement, the configuration used, together with the mesh information, is archived at doi.org/10.5281/zenodo.831485. Mesh partitioning in FESOM is based on a METIS Version 5.1.0 package developed at the Department of Computer Science and Engineering at the





University of Minnesota (http://glaros.dtc.umn.edu/gkhome/ views/metis). METIS and the solver pARMS (*Li et al.*, 2003) present separate libraries which are freely available subject to their licenses. The Polar Science Center Hydrographic Climatology (*Steele et al.*, 2001) used for model initialization and the CORE-II atmospheric forcing data (*Large and Yeager*, 2009) are freely available online. The simulation results can be obtained from the authors upon request.

5   *Competing interests.*   The authors declare that they have no conflict of interest.

*Acknowledgements.*   The public availability of different observational data sets and reanalysis data used in this work is a great help for model development, so the efforts of respective working groups are appreciated. We would like to thank I. Polyakov and B. Rabe for providing us their data compiled from large data sets. Q. Wang is funded by the Helmholtz Climate Initiative REKLIM (Regional Climate Change) project. C. Wekerle is funded by the FRontiers in Arctic marine Monitoring program (FRAM). The model simulations were performed at the

10  North-German Supercomputing Alliance (HLRN).





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
