# Peer review of "A 4.5 km resolution Arctic Ocean simulation with the global multi-resolution model FESOM 1.4"

_Geoscientific Model Development, 2017_

## Referee Comment (RC1) · Dr. Dupont (Referee) · 14 Sep 2017

Recommendation: minor revision

General comments:

in general well written although with some ill-posed expressions at times (see details below). Not groundbreaking but offers some insight into the state-of-the-art Arctic ice-ocean modelling. The unstructured model used here, FESOM, displays a good level of maturity enabling the authors to run sensitivity experiments with different resolutions at an acceptable cost and no particular negative impact on the scientific goals. Thus,

the authors conclude that the higher resolution model is indeed capable of resolving smaller topographic-related details and better the intrinsic variability than the coarser version. In particular they emphasize the role of the Canadian Arctic Archipelago throughflow in impacting the Arctic-wide freshwater pathways. Below I even suggest further comparison between the 3 runs presented.

Detailed comments:

1-page 1, line 17: "freshwater" here sounds awkward. Why not just "precipitation"?

2-page 2, line 14: "[...] the changes are further accelerated by processes of Arctic amplification" does not tell anything. Please elaborate or drop.

3-page 4, line 6: "As the first baroclinic Rossby radius is very small in the Arctic Ocean (Nurser and Bacon, 2014) [...]" Please amend. "very small" is not very telling but I assume the authors mean <5km. Then this statement is only true in the shallow parts of the Arctic and around the GIN seas. It also seemingly contradicts the authors goal to nearly resolve the first Rossby radius in the deep parts of the Arctic where it is about 10km or more with at least 2 points.

4-page 5, line 17: I am not sure what the author meant by "practically optimal": "Almost optimal" or "practical (useful) and optimal"?

5-page 6, ilne 9: "looses" -> "loses"

6-It would probably be telling if the authors could map an instantaneous field for high-lighting the model capacity to (nearly?) resolve mesoscale activity where resolved.

7-Fig 5: please show exact boundaries for domain averaging

8-page 12, line 4: "Different from" sounds awkward. "Contrary to"?

9-Fig10: Given the success of the CAA run to reproduce the same FW pathways as HIGH, I am curious to understand if the CAA run reproduces HIGH in other aspects: profiles, AW layer, SSH... It may be that the eddy-resolving resolution in the deep

ocean is not necessary after all, only a realistic throughflow of the CAA (the eddy parametrization seemingly providing sufficient physics for the rest)!

10-Fig.12 FWC anomaly relative to which period?

11-page 14, line 8: definition of FWC from manuscript: "defined as the amount of pure FW that could be taken out of the upper ocean so that the ocean salinity is changed to 34.8[...]". Just for clarity could you provide the exact depth that defines the upper ocean in your calculation of FWC?

12-page 22, line 17: "2D FW content". why 2D here? FWC is assumed implicitly to be a vertical integral. "maps" maybe?

13-Fig 14, maybe a little outside the scope, but given the pattern of thick ice, I suspect that the ice velocity are too slow. Have the authors compared their sea-ice velocity against buoys or derived-sattelite products?

14-page 26, last paragraph. Can the authors comment on the spurious diffusion on LOW. What are the value of the explicit horizontal diffusion in both simulations? For that matter, it would be nice to have background vertical diffusion value as well...

15-page 28, line 28: "obtains" sounds ill-chosen in this context. "displays" instead?

16-page 30, line 25: "Practically" is again a bit ambiguous. "For practical reasons" maybe?

17-page 30, line 30: "Besides, maintaining high resolution measurements of ocean transports is of great importance for model development too." switch to observation-related subject a bit brutal to the reader. Maybe elaborate a bit?
* * *

---

## Short Comment (SC1) · 11 Oct 2017

Dear Frederic,

Thanks for the encouraging comments. We are still waiting for another review and will update the changes to the paper according to your comments together.

In terms of the sensitivity experiment with refinement only inside CAA, we do learn that it is important to have adequate ocean transport through these narrow straits as mentioned in the paper. However, other aspects, including the Atlantic Water circulation and the spatial distribution of freshwater (especially in the Beaufort Gyre region) still

require good resolution in the Arctic basins. We will make this clear during the revision to avoid confusion.

Best wishes Qiang

---

## Referee Comment (RC2) · Anonymous Referee #2 · 8 Dec 2017

Recommendation: major revision

**General comments**

This manuscript presents an update on the ability of the global, multi-resolution model FESOM1.4 to accuracy simulate the Arctic Ocean. As the authors correctly highlight, despite its size, the Arctic Ocean plays a critical role in the global climate system. The uncertainty in prediction in the Arctic region is a major contributor to uncertainty in global forecasts. This point is highlighted in both the recent Fifth Assessment Report (AR5) from the Intergovernmental Panel on Climate Change (IPCC) and the Coupled Model Intercomparison Project Phase 5 (CMIP5) of the World Climate Research Pro-

gramme (WCRP). Efforts to improve the accuracy of numerical modelling in the region, and therefore of future forecasts, are critical at this time. I applaud the authors for taking up this challenge in exploring how this can be achieved.

It is an impressive feat that an unstructured mesh model of FESOM's nature is now taking part in the international model intercomparison CORE-II. It is fundamentally quite a different approach to the majority of models in CORE-II (and the complimentary CMIP study), and as a consequence, offers opportunities to improve process modelling that is not possible in other cases. In this paper the authors look at increasing horizontal spatial resolution in the Arctic region. It should then be possible to resolve smaller-scale processes, and overall, a larger range of processes – which has the potential to improve overall prediction of key properties such as water mass and heat transport, and sea-ice migration. Intuitively, this is a logical step – that ncreasing resolution in regions of the globe where there is largest uncertainty will improve prediction. This is however, certainly not a given result, which makes it a very worthy area of study.

The initialisation of flexible, unstructured mesh models such as FESOM is significantly more complex than the structured models that are traditionally used in climate modelling studies. With variable resolution possible, the question on the best spatial discretisation to use is a huge question. This is very much an ongoing area of research and one which would be very difficult to address definitively in a single paper such as this. So whilst it is somewhat unsatisfactory to take a single mesh choice, run the coupled model in this configuration and analyse the output – there is still the potential to learn significantly from this type of exercise. It is interesting the authors additionally include an intermediate case with relatively high resolution restricted to the channels of the Canadian Arctic Archipelago (CAA). As the authors highlight, treatment of the CAA channels varies significantly between climate models because they are difficult to resolve with the fixed resolutions in the majority of climate models. As the authors find, the feedback from resolution on uncertainty is heterogeneous. It is very useful this is included in the paper, and I expect could be the subject of a separate work given the

Interactive
comment

options that could be explored, and comparisons made to existing approaches.

**Specific comments**

1. *Focus.* As stated, the research is very timely and I am pleased to see this being covered and particularly in a global context. I think the basis of the research idea here is excellent. As far as I can read into the execution and analysis, this too appears excellent. Saying this, I found the write up difficult to follow. It is very long and feels like it wants to act as: (a) a review of physical processes in the Arctic, (b) introduce a multi-resolution global model with a high resolution Arctic, whilst also (c) evaluating the multi-scale model. I understand it is a complex system, and there is some need to discuss expected physics, but as a GMD model evaluation paper, I think it would help the reader significantly if focus is directed at (c). Details of (b) to be included as required for reproducibility (since the model 1.4 is introduced in past papers) and (a) discussed for context of the analysis of model output.

   Related to (a), some paragraphs go into a detailed description of physical processes in the ocean, only to end with a general suggestion that more work is required. It is good to include a description of the physics, but only if this is relevant to specific analysis of model output, or in direct relation to the basis of this work – with increased resolution.

   Cleaning up these and the more general statements that appear but seem unnecessary and do not add to the paper – would help focus the paper, reduce verbosity and make it more accessible.

   Some examples in the introduction:

   – *"Numerical modeling can be used to understand the dynamics of the ocean and predict its future changes."*

- *"Large model biases in the upper Arctic Ocean are another common issue in many ocean general circulation models."* – which models? and links to why? Is this a resolution issue? These statements need to be specific about models and approaches when this paper is exactly about a different modelling approach that could solve these issues.

- *"Model simulation results can be sensitive to model configuration, including the choice of numerical and physical schemes, parameters and grid resolution."* – surely model simulation results *are* directly dependent on model configuration?

- *"As computational resources grow with time, the modeling community tends to use higher and higher model resolution. Certainly there is a need in the modeling community to evaluate high resolution models with respect to the common model issues identified in previous model studies."* – again this is very verbose, is not specific and does not add anything significant. Which *"common model issues"*? Which *"previous model studies"*?

- *"These studies provided background knowledge on the Arctic Ocean representation in those models and identified their common issues."* – again a weak summary of what has already been said previously. This repetition with no specifics is not helpful.

General sentences like these should be removed.

Other examples of superfluous content:

- Page 5, line 12: *"It helps to preserve monotonicity and eliminate overshoots. When compared to a second order scheme without flux limiter and an implicit second order scheme in idealized 2-D test cases, at coarse resolution this FCT scheme tends to slightly reduce local maxima even for a smooth field, but at high resolution it well represents sharp fronts and shows least dispersion errors (Wang, 2007)."* This is a model detail under (b), but it is not

clear how this is relevant to the focus of the paper. Please either elaborate on how this is relevant or remove.

– This is again repeated on page 31, line 32, and does not appear necessary to include. Here a link is at least suggested to the work, *"which can explain the obtained improvement of the AW layer in the high resolution simulation"* – but is only loose conjecture. This is something that could be tested here in this modelling framework. Please either test and include or remove.

– The following paragraph (beginning *"The diapycnal mixing is.."*) could also be potentially removed. These are useful details of the model setup, but standard for FESOM1.4, and all detailed in the model description paper Wang et al. (2014). If these choices are different to others (in CORE-II, for example) and pertinent to the success of the model here, please elaborate (as per the later on SSS restoring), otherwise remove.

– *"Besides identifying the impact of horizontal resolution on the Arctic Ocean main circulation, we also discussed scientific questions and model issues that need to be explored in future work, and some of the illustrated model issues are common in many other ocean-sea ice models. Overall, increasing model resolution does considerably improve the performance of the Arctic Ocean simulation, while further efforts are necessary to solve remaining issues that are not linked to applied model resolution, and to develop/improve parameterizations that are still required even with best resolution affordable now."* – this lengthy paragraph adds very little except a very general comment that it has been shown that increased resolution helps. Please remove.

In general, please ensure all discussion of physical processes directly links to quantitative analysis of model performance and output. Detailed descriptions of physical process in the Arctic that end with loose general statements that these could be important / need to be considered when modelling, do not add significantly to the paper and its aim to evaluate the 3 configurations. The increased

length only makes the paper more inaccessible.

2. *Structure*. This paper is part of a concerted effort to develop FESOM, improve estimation in the Arctic and this is the subject of many recent related papers. (The lead author is lead author on 3 cited works in 2016 alone). Given this, the paper would greatly benefit from a focused "positioning" section.

   Before jumping straight into *"Model setup"* I suggest a section briefly reviewing the effort to date, how each of the relevant papers fits in and solves/identifies lines of research to arrive at the work here. Some of this can be collected from snippets spread throughout the paper – including for example parts of the introduction, and a large part of *"5.3 Unstructured-mesh modeling"* (which I found odd positioned at the end of the paper).

   On the introduction, the beginning 3 paragraphs are good and give a nice motivation for the work. The fourth paragraph jumps straight into the aim of the paper to investigate resolution – which at that point is not motivated – but then jumps back into motivation discussing the narrow straits. The fifth paragraph then jumps back out to talk about the overall modelling framework in general development. I suggest the fourth paragraph discusses the narrow straits, to then motivate and justify why an increase in resolution could help. There is some good justification given in Section 5.3 (e.g. *"An adequate representation of the CAA throughflow is found to be very important. With an unstructured-mesh model like FESOM, one can locally increase model resolution to accurately resolve the narrow channels and faithfully simulate the FW export (Wekerle et al., 2013). . ."*, and more therein). There is also a review of higher resolution modelling efforts there that is appropriate for the motivation in the introduction, and to give context to this approach.

   This then naturally leads into a short paragraph outlining the aims here – bringing in the 4.5km high resolution, potentially addressing issues prevalent in CORE-II.

Please make this concise. For example, the following, all in the latter part of the introduction, largely say the same:

– *"In this paper we will evaluate a high resolution Arctic Ocean simulation and ellucidate the sensitivity of model results to resolution."*

– *"As one of the first steps towards designing such a system, in this paper we evaluate the simulated Arctic Ocean by FESOM on a global mesh with 4.5 km resolution inside the Arctic Ocean".*

– *"We will compare the 4.5 km model results with those from this coarse setup to understand the impace of model resolution."* (note this typo – I think should be *"impact"*)

Then I suggest the "positioning" section to make it clear what has been achieved with this approach, related studies (e.g. the 4.5km sea-ice modelling study *"Werkerle et al. (2013) The Canadian Arctic Archipelago throughflow in a multiresolution global model: Model assessment and the driving mechanism of interannual variability"* and how this studies fits / differs / is an important step in this continuing development.

3. *Figure 1*. This is a good figure and helpful to include for reference. It is also useful to locate the vertical section in figure 8. I suggest the *"curves"* are better identified as *"arrows"*, since they are orientated markings. In hardcopy, it was difficult to make out the blue arrows and blue text *"Transpolar drift"* on top of the scalar blue background. I suggest different colours are used to make a clear distinction between the arrows and text vs. the background. Possibly the background scalar depth field could be coloured in green? The arrows outlined in white?

4. *Figure projections*. Figures 2(a) and (b) are compared – and it is good to discuss resolution of the Rossby radius in this way – but they are presented in different projections, which makes a comparison by eye difficult.

Moreover, Figure 1 is used for orientation – again this is helpful – but it is orientated differently to 2(b) and the subsequent 4, 6 7, 10, 11, 14 and 15. Please make the orientation consistent, such that Figure 1 can be used to easily identify regions when later output figures are discussed.

5. *Definition of HIGH and LOW*. The introduction refers to Figure 2 on page 4. Figure 4, on page 5, refers to HIGH, but this is not defined until later in the text on page 6. Please fix so the reader knows what HIGH refers to before it is used in text.

6. *Eddy resolving considerations*. Figure 2 and the connected text make an analysis of where the fixed mesh can resolve mesoscale eddy activity. This quantitative analysis, including the spatial dependence presented in 2(b), is then significantly undermined by the Figure 2 caption comment: *"Note that effective model resolution usually is coarser than the grid size due to numerical dissipation."*

Please change this into a quantitative argument. As it stands the reader really is at a loose end in knowing where the model under consideration is mesoscale eddy resolving. This point is part of the discussion, and the reason for figure 2(b) – and link to / motivation for the increased resolution – so it is important it is completed. Is there a quantitative study of FESOM1.4 (or the underlying discretisation/numerical dissipation present) that can be used to infer the effective model resolution, and where in fact, in this configuration being evaluated here, the model is mesoscale eddy resolving?

7. *Reproducibility and Zenodo archive*. The authors have provided a DOI link to a Zenodo archive. I was able to access the link at `https://doi.org/10.5281/zenodo.831484` and download the archive `fesom1.4.tar.bz2`. After attempting various methods, unfortunately I could not access the contents of the archive. For example, trying:
`tar tvjf fesom1.4.tar.bz2`

`tar: This does not look like a tar archive`
```
tar: Skipping to next header
tar: Exiting with failure status due to previous errors
```

Please take a look at the archive. It appears the format is not as expected. Please provide instructions on how to access the data. Alternatively, and a better solution in my opinion, I suggest uploading files as they are, and not compressed or contained in an archive (Zenodo can handle large single files <50GB) – to avoid problems in opening/uncompressing files.

8. *Mesh set-up and reproducibility*. A large focus of the paper is on the three mesh configurations, LOW, HIGH and high only in the CAA. Sometimes the labels *"24 km"* and *"4.5km"* are used for LOW and HIGH in the figures. It could be helpful to keep this consistently to the defined labels LOW and HIGH. For the third mesh configuration with high resolution only in the CAA, various labels are used. *"CAA HIGH"* appears in figure 10 but is not defined. The graphic uses *"CAA 4.5 km"*. It would be helpful to stick to one label for all throughout. Please define a clear label for this third case, and ideally together with the definitions of the LOW and HIGH cases. It would also be helpful to make it clear in this same place early on that there are 3 different mesh configurations considered here.

The description of how the resolution varies and in particular how it varies between the three is not described sufficiently. This makes it difficult to reproduce or even compare results. Another flexible mesh model may wish to use 4.5km resolution in the Arctic in the same way to compare and contrast. It is not clear how this would be achieved without significant ambiguity over how resolution is varied, bar the broad, general description *"On the second mesh (HIGH) the horizontal resolution is further increased to 4.5 km inside the Arctic Ocean (defined by the Arctic gateways of Bering Strait, CAA, Fram Strait and Barents Sea Opening, Fig. 2a)"* and the image in figure 2a (which would be difficult to extract this data from and also includes only half the globe). This paper is evaluating the

sensitivity of output to mesh resolution. It is important this is characterised fully. In what way does the resolution vary from 4.5km to 24km in mesh HIGH? The 4.5km region is defined as the Arctic Ocean, closed with reference to the gateways. Yet this description appears incomplete, since in figure 2a it appears higher resolution is applied in the Baltic Sea also?

On reproducibility, can the bounds of the 4.5km region used in this case be defined by a mathematical function, in terms of longitude and latitude for this purpose? If not, can shapefile definitions be provided? Other multi-resolution models may not use a triangular mesh, and require accurate description of regions for generation. Please provide sufficient information such that the 3 three mesh configurations can be regenerated – in storage that is persistent and can be reliably depended on, ideally with a DOI.

The description of the third configuration is even less well-defined: *"It has resolution similar to LOW outside CAA, but the same resolution as in HIGH inside CAA."* and leaves the reader questioning why is it not the same resolution as LOW outside the CAA? How is the CAA region defined? How does the resolution vary as you move to outside of the CAA?

This is an important study that others will want to compare to. It is important the 3 configurations can be regenerated. Also, ideally there will be sufficient information that others would be able to re-run the same FESOM simulations.

The authors advocate a model intercomparison using high resolution cases. It seems this would be facilitated with sharing as much as possible of the model configurations and model output. Why not make the 3 mesh configurations available on a persistent resource such as Zenodo? It would also benefit a model intercomparison if key outputs from FESOM1.4 here were also made available in this way – for others to contrast and compare.

9. *Spin up and simulation time frames.* The spin up part of model runs is repeatedly

referred to throughout, but it is not clear how long this is and what forcing data is used. Section 4.2 mentions it takes nearly 30 years for salinity to reach a quasi-equilibrium state. Is this enough for the whole model? In the section on freshwater 5.2, 20 to 30 years is mentioned for a specific sensitivity run exploring the effect of SSS restoring. Please clarify.

Page 6, line 29 implies the 3 configurations are run from 1950 to 2009 (with some spin up assumed beforehand, relaxing to a set climatology?). This is supported by figures such as 3, 8 and 9 with time axes over this period.

Figure 4 shows an average from 1970-1999, figure 8 from 1980-1999, figure 6 compares the 70s to 90s, figure 7 is not clear on range considered. Figures 10 and 11 considers different periods 1993-2002, 2003-2007 and 1996-2009, and figure 14 and 15 other ranges. In some cases these choices are justified, but others not. Please make sure it is clear why particular time ranges have been chosen. For example, why in figure 8 miss the contribution from 1970-1979 which is included in figure 4? Indeed why not an average of the entire simulated record over the CORE-II time range?

For a paper on model evaluation and more so when differing spatial resolutions (and by CFL, time stepping) are involved, one would expect an analysis of model performance with respect to time. This is touched on with a loose reference to 7 days throughput per day. Please include details of time steps used in all three configurations and simulation time per time step on each mesh. If possible include a scaling plot. It would also be interesting to compare simulation time per time step for each of the model components (e.g. ocean, sea-ice) on each mesh configuration. This information is very important for comparison studies, to guide choices in the development of other models and for users in model choice – possibly even more so here given FESOM2.0 is being developed with a different dynamical core and discretisation choice.

On *"The deficiency indicates a clear requirement for eddy resolving resolution*
*in the Fram Strait region in order to faithfully simulate the amount and property of AW that enters the Arctic basins through the Fram Strait. In ocean climate simulations, however, it is hardly possible to afford 1 km model resolution in the near future."* (page 28, line 3): it would be helpful to be more quantitative here – how much computational effort do the authors believe a 1km version of the model would require? Analysis in Holt et al. (GMD, 2017) implies it will be possible to run global models more routinely down to coastal scales of 1.5km in the next 10 years.

The following *"Accordingly, efforts on parameterizations are required to improve the simulation of AW circulation in Fram Strait."* does not add much. Please add clarification on what parameterisations are required or remove.

Page 30, line 21: *"However, if the finest grid size is used in narrow straits in FE-SOM, the model time step and the overall model throughput could be constrained by this grid size."* What else is it likely to be constrained by? It seems very likely rather than *"could"*. A breakdown of costs of model components (e.g. ocean, sea-ice, ...), including the cost impact of moving to 4.5km in the Arctic region or just to the CAA – suggested above – would help give a quantitative answer here. As it stands the reader has no feeling for what the next step might be. What is the computational impact of HIGH vs HIGH CAA? – an important aspect of evaluating these model configurations.

10. *Atlantic Water core temperature prediction*.

Examining Figure 4(a), it appears the LOW model over-estimates the Atlantic Water core temperature (AWCT) compared to PHC climatology. Moving to HIGH this over-estimation increases significantly. Notably warmer waters (of over 1 degree above the PHC climatology) appear to propagate through the Fram Strait into the Arctic Ocean region. In this regard, AWCT temperature \*distribution\* appears to worsen in the move to using high resolution. Please comment.

How do instantaneous distributions appear? A significant error is seen in HIGH's LFW transport anomaly across the Fram Strait between 1950-1970 (not apparent in LOW) – is this linked? (although Figure 4 contains model results averaged 1970-1999).

On discussing heat content, with both LOW and HIGH integrated content higher, further analysis is dismissed with the following: *"Due to inaccuracy in diagnosing heat budget terms (e.g., caused by interpolation) and ignoring heat diffusion in model output, the mismatch between the ocean heat content changing rate and Arctic net heat flux can have the same order of magnitude as this value."* (page 26). It is not clear what is meant here. Please reword and improve the explanation. It does not seem satisfactory that the increased heat content cannot be explained.

*"Our model results show that the magnitude of the AW temperature is not lower in LOW"* (page 26) – is this shown in Figure 5? Is this the maximum or depth-integrated magnitude of AW temperature?

11. *Over-estimation of liquid freshwater content.* Both LOW and HIGH over-estimate the liquid freshwater (FW) content. This is noted on page 18, line 11. What are the reasons for this over-estimation? Is it expected this can be further reduced by further increasing resolution?

*"The variety of FW content distributions simulated in different ocean models shown by Wang et al. (2016b) presumably can be partly attributed to different model representations of the CAA region."* – this is could be true, but this does not add significantly. Can it be backed up by other studies? What are the different representations? How do they relate to the HIGH CAA configuration here? Do other representations – that are most closely related to the one here – show similar changes/improvements/indications? Some of this is discussed much later on in 5.3 and 6. It appears a key consideration here given focus on resolution and representation and might be good to have its own section?

12. *Passive tracer implementation*. Figure 11 includes the comments *"Note that the passive tracers were set to zero south of Fram and Davis Straits. The passive tracers are averaged over the upper 100 m."* These are details better included inline in the text. This raises additional questions. For the tracer $\phi$, the above implies:

$$\frac{\partial}{\partial t} \int_V \phi \, dV \neq 0 \qquad (1)$$

So tracer total is not conserved? Please elaborate on why this is done and its implications.

What is the reason for averaging the tracer over the top 100m?

Is this type of analysis done elsewhere, under the same conditions?

13. *FW variability*. Referring to these sentences in section 4.4:

– *"Compared to the period of 1980-2000, the mean Arctic FW content averaged over the 2000s has increased by about 4500 km3 based on observations (Polyakov et al., 2013b; Haine et al., 2015), while the increase is only about 1700 km3 in our two simulations (Table 1)."*

– *"On average the 13 CORE-II models analyzed in Wang et al. (2016b) underestimated the observed upward trend also by half."*

Observations show mean Arctic FW content has increased nearly 3 times that seen in the model simulations presented. Why is this? The paper highlights that on average CORE-II models underestimate by a half. These models have a significantly lower resolution than the HIGH case considered here, yet the above implies this higher-resolution simulation performs more poorly than the average. Why is this?

The paragraph on temporal variation (page 22, line 7) ends with *"Simulation HIGH consistently obtains positive changes in the Eurasian Basin, but with a larger*

*magnitude. It has negative values north of Greenland, which is not present in the observation."* which appear unexplained. Can you suggest why negative changes are seen north of Greenland? and the positive change in the Eurasian Basin?

On *"The interannual variability of FW transport through the Arctic gateways shows large similarity between the two simulations (Fig. 13a-c)."* It does not seem correct to characterise this as a large similarity. There is some agreement in the anomaly trend, but also large deviations. Fig. 13(a) shows HIGH has a very significant deviation in the first two decades 1950-1970 of the six decades considered. The magnitude of the deviation is of the order of the max change seen, so arguably significant. Why the deviation? Is it linked to the excessively warmer waters see entering through the Fram Strait in the model output?

14. *Outcomes.* *"Instead, we often modify the geometry of the CAA channels to allow adequate CAA throughflow. Such model adjustment, however, is not trivial as shown by the large model spread in CAA FW transports among the ocean climate models analyzed in Wang et al. (2016b)."* (Page 30, line 27). This can be considered yet another parameterisation and the same approaches made to analyse its impact.

*"When developing global climate models, the modeling groups certainly need more efforts to better adjust the 30 CAA representation. Besides, maintaining high resolution measurements of ocean transports is of great importance for model development too."* (Page 30, line 29). This does not add anything significant. Can the authors suggest a solution to CAA representation/parameterisation?

*"Most of the models analyzed in past CORE-II model intercomparison studies have relatively coarse resolution. For developing our unstructured-mesh model system with regional focus, it would be helpful to communicate experience with the large structured-mesh model community, for example, in future high resolution ocean climate model intercomparison projects."* (Page 31, line 1). This

could be better worded. Do you mean *"Most of the test cases/configurations in past CORE-II..."*? – and you are suggesting a suite of higher resolution inter-comparison test cases?

15. *Other small points*.

   (a) Last line of the abstract sounds disjoint, given the paper concerns increased resolution. Maybe something like: *"Along with increased resolution, we additionally discuss other issues that could benefit from development to help increase accuracy in the region, including the improvement of parameterizations, for example."*

   (b) Lars Smedsrud et al. have recently published (January 2017 – you cite as *"Smedsrud et al., 2017"*) on Fram Strait sea ice export – can this help update the 11+ year old figure and error in Table 1? "Fram Strait sea ice export variability and September Arctic sea ice extent over the last 80 years" The Cryosphere, doi:10.5194/tc-11-65-2017. There appears to be some large deviations in Fram Strait fluxes here, comparing FESOM model output and observations.

   (c) Page 24, line 21: Part of a general request for more quantitative statements: *"Note that much higher model resolution is required in order to simulate sea ice leads with realistic width."* What resolution is required? Is it therefore expected that increase in sea-ice model resolution which see no advantage until this is reached?

**Technical corrections**

1. Use of the definite article "the" appears to have been skipped in multiple places throughout the paper. e.g. Page 10, line 6: *"The maximum temperature in Eurasian Basin in simulation"*, Page 11, line 8: *"correct circulation direction in Canadian Basin"*.

2. Page 1, line 14: *"including improving parameterizations"*, better: *"including the improvement of parametertizations"*.

3. Page 1, line 22: *"deep water formation regions"* better worded as *"regions where deep water is formed"*?

4. Page 2, line 16: *"lower latitudes ocean and climate"*, change to *"lower latitude ocean and climate"*.

5. Page 2, line 16: *"societal"* is better here than *"social"*.

6. Page 2, line 16: *"that remains under debate"* better.

7. Please check capitalisation in references, e.g.

   – Wekerle et al. (2013) should contain *"Canadian Arctic Archipelago"*.
   – Schauer et al (2002) *"Amundsen"* and *"Makarov"*.
   – *"Arctic"*, *"Fram"* and *"Barents"* in Maslowski et al. (2004).
   – ...

8. Page 25, line 3: *"(e.g., Smedsrud"* comma not needed – please check throughout.

9. Figure 14(f) is not labelled *"(f)"*. Also, the coastline is not identified like the others (a)–(e).

10. Figure 2 caption: *"Values larger than one indicate mesoscale eddy resolving."* – the meaning is understood, but please make into a sentence.

11. Page 5, line 3: *"It work with"*, change to *"It works with"*.

12. Page 5, line 15: *"shows \*the\* least"*.

13. Page 6, line 21: *"one over the ocean column"* – is this one over the ocean depth? water column thickness?

14. Pages 4 and 6: Please avoid repetition of the *"Time frame of 7 model years per day"*.

15. Min and max values are missing in many colour bars – e.g. Figures 2b, 3a, 3c, 7a, 7b, 8a-e, 10, ... – This is very helpful for future studies comparing to this work. Please ensure all min and maxes are labelled.

16. Some axis do not have end values – e.g. time axes in Figure 9. Is this data until 2009? Please make it clear where the axis ends (mark with year, or make it a decade to 2010). Same with Fig 3, 12, 13, ...

17. Use of the definite article "the" appears inconsistent – e.g. *"towards the Fram Strait"* on page 7 (twice), but *"toward Fram Strait"* on page 8.

18. Spelling of *"elucidate"* page 2, line 34.

19. *"Dupont, F., and others (2015)"* – please elucidate on the *"others"*.

20. *"CAA"* unnecessarily redefined in figure 10.

21. Possibly move the reference to Holloway et al. (2007) on page 11 up to the introduction of topostrophy – so readers know where to find its definition.

22. Page 11, line 10: *"Indeed, the Arctic Ocean hydrography obtained on mesh LOW was found to be one of the well simulated when comparing the state-of-the-art ocean climate models (Ilicak et al., 2016)"* – *"one of the well simulated"* does not read well and the sentence is verbose. Last part better rephrased as *"when compared to the suite of state-of-the-art climate models in Ilicak et al. (2016)"* ?

23. Page 12, line 10: *"between the two models"* or *"between models"*.

24. Page 18. line 2: Please refer directly to the section number rather than this loose reference.

25. Page 24, line 11: *"in \*the\* 2000s"*.

26. Page 30, line 4: *"without necessity"*, please reword – *"without \*the\* necessity"*?

27. Page 33, line 1: Space in URL.

---

## Author Comment (AC1) · 20 Dec 2017

Dear Dr. Dupont,

Thank you very much for your helpful comments. We did the revision according to your comments. The reply details are given below.

1-page 1, line 17: "freshwater" here sounds awkward. Why not just "precipitation"?
**reply**: Changed to "precipitation" (p1, l17)

2-page 2, line 14: "[...] the changes are further accelerated by processes of Arctic amplification" does not tell anything. Please elaborate or drop.
**reply**: Dropped.

3-page 4, line 6: "As the first baroclinic Rossby radius is very small in the Arctic Ocean (Nurser and Bacon, 2014) [...]" Please amend. "very small" is not very telling but I assume the authors mean <5km. Then this statement is only true in the shallow parts of the Arctic and around the GIN seas. It also seemingly contradicts the authors goal to nearly resolve the first Rossby radius in the deep parts of the Arctic where it is about 10km or more with at least 2 points.
**reply**: the range of Rossby radius is added (p4, l3)

4-page 4, line 17: I am not sure what the author meant by "practically optimal": "Almost optimal" or "practical (useful) and optimal"?
**reply**: Changed to „In practice, however, optimal ..." (p4, l14)

5-page 7, line 9: "looses" -> "loses"
**reply**: changed (p7, l22)

6-It would probably be telling if the authors could map an instantaneous field for high- lighting the model capacity to (nearly?) resolve mesoscale activity where resolved.
**Reply**: We do not consider this model setup as well eddy-resolving with which one can focus on mesoscale processes.

7-Fig 5: please show exact boundaries for domain averaging
**reply**: Definition is added in the text (p8 l15): „The two basins are defined as the Arctic region where the ocean bottom is deeper than 500 m, and separated by the Lomonosov Ridge."

8-page 12, line 4: "Different from" sounds awkward. "Contrary to"?
**reply**: Changed. (p12, l22)

9-Fig10: Given the success of the CAA run to reproduce the same FW pathways as HIGH, I am curious to understand if the CAA run reproduces HIGH in other aspects: profiles, AW layer, SSH... It may be that the eddy-resolving resolution in the deep ocean is not necessary after all, only a realistic throughflow of the CAA (the eddy parametrization seemingly providing sufficient physics for the rest)!
**Reply**: We had mentioned this aspect in the second last paragraph of the paper. In the revision we added one sentence, „We also found that only better resolving the CAA channels (in the simulation where only the CAA is resolved with 4.5 km) did not significantly impact the representation of the AW layer." This additional sensitivity experiment shows the impact of CAA on FW spatial distribution, but not on AW circulation. Further extended studies, if required, can only be done in separate work considering the current paper length. (p33, l5)

10-Fig.12 FWC anomaly relative to which period?
**Reply**: Relative to the mean of the plotted period. Added to the figure caption.

11-page 14, line 8: definition of FWC from manuscript: "defined as the amount of pure FW that could be taken out of the upper ocean so that the ocean salinity is changed to 34.8[...]". Just for clarity could you provide the exact depth that defines the upper ocean in your calculation of FWC?
**Reply**: We clarified it now: „In the calculation of the modelled FW content presented below, the integration is taken from ocean surface to the depth where salinity is equal to the reference salinity" (p14, l8)

12-page 22, line 17: "2D FW content". why 2D here? FWC is assumed implicitly to be a vertical integral. "maps" maybe?
**Reply**: changed to „vertically integrated" (p22, l13)

13-Fig 14, maybe a little outside the scope, but given the pattern of thick ice, I suspect that the ice velocity are too slow. Have the authors compared their sea-ice velocity against buoys or derived-satellite products?
**Reply**: We checked the simulated sea ice velocity. On the contrary, the simulated sea ice drift is higher than observed from satellites. We consulted colleagues working on observations, and they suggested that the observation products might underestimate the drift was well. Further efforts are required in understanding the behavior of sea ice velocity, both in the model and in the observation.

14-page 26, last paragraph. Can the authors comment on the spurious diffusion on LOW. What are the value of the explicit horizontal diffusion in both simulations? For that matter, it would be nice to have background vertical diffusion value as well...
**Reply**: We did not quantify the spurious numerical mixing coefficients. However, we performed sensitivity simulations, and found that changing the background vertical diffusivity on the order of 1e-6 m²/s can significantly change the model results. This indicates that numerical mixing, even small, can still significantly impact model results. Lateral mixing coefficients were mentioned in the model setup section. The importance of vertical mixing coefficients was mentioned in line 10 on page 29. The value of background vertical mixing coefficient is added (line 13 on page 6).

15-page 28, line 28: "obtains" sounds ill-chosen in this context. "displays" instead
**reply**: Changed to „has"

16-page 30, line 25: "Practically" is again a bit ambiguous. "For practical reasons" maybe?
**reply**: Changed to „in practice" (p31, l7)

17-page 30, line 30: "Besides, maintaining high resolution measurements of ocean transports is of great importance for model development too." switch to observation-related subject a bit brutal to the reader. Maybe elaborate a bit?
**reply**: The sentence is removed.

Best regards
the authors

---

## Author Comment (AC2) · 20 Dec 2017

Dear reviewer,

Thank you very much for your helpful comments. We did the revision according to your suggestions and the detailed reply is listed below.

Specific comments

1. Focus.
As stated, the research is very timely and I am pleased to see this being covered and particularly in a global context. I think the basis of the research idea here is excellent. As far as I can read into the execution and analysis, this too appears excellent. Saying this, I found the write up difficult to follow. It is very long and feels like it wants to act as: (a) a review of physical processes in the Arctic, (b) introduce a multi-resolution global model with a high resolution Arctic, whilst also (c) evaluating the multi-scale model. I understand it is a complex system, and there is some need to discuss expected physics, but as a GMD model evaluation paper, I think it would help the reader significantly if focus is directed at (c). Details of (b) to be included as required for reproducibility (since the model 1.4 is introduced in past papers) and (a) discussed for context of the analysis of model output. Related to (a), some paragraphs go into a detailed description of physical processes in the ocean, only to end with a general suggestion that more work is required. It is good to include a description of the physics, but only if this is relevant to specific analysis of model output, or in direct relation to the basis of this work – with increased resolution. Cleaning up these and the more general statements that appear but seem unnecessary and do not add to the paper – would help focus the paper, reduce verbosity and make it more accessible. Some examples in the introduction:

–

"Numerical modeling can be used to understand the dynamics of the ocean and predict its future changes."
**Reply**: removed.

–

"Large model biases in the upper Arctic Ocean are another common issue in many ocean general circulation models." which models? and links to why? Is this a resolution issue? These statements need to be specific about models and approaches when this paper is exactly about a different modelling approach that could solve these issues.
**Reply**: citation is added to this sentence. (p2 l25) The sentences after it explains details.

–

"Model simulation results can be sensitive to model configuration, including the choice of numerical and physical schemes, parameters and grid resolution."
surely model simulation results *are* directly dependent on model configuration?
**Reply**: sentence removed.

–

"As computational resources grow with time, the modeling community tends to use higher and higher model resolution. Certainly there is a need in the modeling community to evaluate high resolution models with respect to the common model issues identified in previous model studies."
Again this is very verbose, is not specific and does not add anything significant. Which "common model issues" Which "previous model studies"
**Reply**: We make this sentence more exact: "Certainly there is a need in the modeling community to evaluate high resolution models with respect to the common model issues identified in previous model studies as mentioned above." (p2, l8)

–

"These studies provided background knowledge on the Arctic Ocean representation in those models and identified their common issues." Agaig a weak summary of what has already been said previously. This repetition with no specifics is not helpful.

**Reply**: "These studies provided background knowledge on the Arctic Ocean representation in those models and identified their common issues." Therefore, "In this paper we will mainly focus on the key diagnostics used in these studies for evaluating our simulations." Because of the experience gained in these studies, we now know what we should focus on in this study. (p4, l25)

General sentences like these should be removed.
Other examples of superfluous content:

– Page 5, line 12:
"It helps to preserve monotonicity and eliminate overshoots. When compared to a second order scheme without flux limiter and an implicit second order scheme in idealized 2-D test cases, at coarse resolution this FCT scheme tends to slightly reduce local maxima even for a smooth field, but at high resolution it well represents sharp fronts and shows least dispersion errors (Wang, 2007)." This is a model detail under (b), but it is not clear how this is relevant to the focus of the paper. Please either elaborate on how this is relevant or remove.
**Reply**: removed.

– This is again repeated on page 31, line 32, and does not appear necessary to include. Here a link is at least suggested to the work, "which can explain the obtained improvement of the AW layer in the high resolution simulation" but is only loose conjecture. This is something that could be tested here in this modelling framework. Please either test and include or remove.
**Reply**: It was shown that the AW layer can be better represented (the thickness in particular) with reduced numerical mixing by Holloway et al. (2007), and it was shown that numerical mixing can be reduced with increasing resolution by Wang (2007) in FESOM idealized experiments. It is then logical to link the improved AW layer with 4.5km resolution to reduced numerical mixing in the model. Here we decided to keep it.

– The following paragraph (beginning "The diapycnal mixing is..") could also be potentially removed. These are useful details of the model setup, but standard for FESOM1.4, and all detailed in the model description paper Wang et al. (2014). If these choices are different to others (in CORE-II, for example) and pertinent to the success of the model here, please elaborate (as per the later on SSS restoring), otherwise remove.
**Reply**: This paragraph provides some of the key information required by modellers. Even though they are the standard settings, we always repeat them when describing the model setups in papers, as readers want to know key parameters directly in the paper they are reading rather than looking for other papers. We certainly did not mention those secondarily important settings in the paper.

–
"Besides identifying the impact of horizontal resolution on the Arctic Ocean main circulation, we also discussed scientific questions and model issues that need to be explored in future work, and some of the illustrated model issues are common in many other ocean-sea ice models. Overall, increasing model resolution does considerably improve the performance of the Arctic Ocean simulation, while further efforts are necessary to solve remaining issues that are not linked to applied model resolution, and to develop/improve parameterizations that are still required even with best resolution affordable now."
This lengthy paragraph adds very little except a very general comment that it has been shown that increased resolution helps. Please remove. In general, please ensure all discussion of physical processes directly links to quantitative analysis of model performance and output. Detailed descriptions of physical process in the Arctic that end with loose general statements that these could be important / need to be considered when modelling, do not add significantly to the paper and its aim to evaluate the 3 configurations. The increased length only makes the paper more inaccessible.

**Reply**: This paragraph is the last one of the whole paper, and we summarized the paper and provided the outlook. In terms of the outlook, we emphasize that many other aspects should be investigated besides increasing model resolution, which is worth addressing at the end. We are inclined to keep this paragraph.

2. Structure

This paper is part of a concerted effort to develop FESOM, improve estimation in the Arctic and this is the subject of many recent related papers. (The lead author is lead author on 3 cited works in 2016 alone). Given this, the paper would greatly benefit from a focused "positioning" section. Before jumping straight into"Model setup" I suggest a section briefly reviewing the effort to date, how each of the relevant papers fits in and solves/identifies lines of research to arrive at the work here. Some of this can be collected from snippets spread throughout the paper – including for example parts of the introduction, and a large part of "5.3 Unstructured-mesh modeling" (which I found odd positioned at the end of the paper). On the introduction, the beginning 3 paragraphs are good and give a nice motivation for the work. The fourth paragraph jumps straight into the aim of the paper to investigate resolution – which at that point is not motivated – but then jumps back into motivation discussing the narrow straits. The fifth paragraph then jumps back out to talk about the overall modelling framework in general development. I suggest the fourth paragraph discusses the narrow straits, to then motivate and justify why an increase in resolution could help. There is some good justification given in Section 5.3 (e.g. "An adequate representation of the CAA throughflow is found to be very important. With an unstructured-mesh model like FESOM, one can locally increase model resolution to accurately resolve the narrow channels and faithfully simulate the FW export (Wekerle et al., 2013) ..., and more therein). There is also a review of higher resolution modelling efforts there that is appropriate for the motivation in the introduction, and to give context to this approach. This then naturally leads into a short paragraph outlining the aims here – bringing in the 4.5km high resolution, potentially addressing issues prevalent in CORE-II. Please make this concise. For example, the following, all in the latter part of the introduction, largely say the same:

– "In this paper we will evaluate a high resolution Arctic Ocean simulation and elucidate the sensitivity of model results to resolution."

– "As one of the first steps towards designing such a system, in this paper we evaluate the simulated Arctic Ocean by FESOM on a global mesh with 4.5 km resolution inside the Arctic Ocean".

– "We will compare the 4.5 km model results with those from this coarse setup to understand the impace of model resolution." (note this typo – I think should be "impact") Then I suggest the "positioning" section to make it clear what has been achieved with this approach, related studies (e.g. the 4.5km sea-ice modelling study"Werkerle et al. (2013) The Canadian Arctic Archipelago throughflow in a multiresolution global model: Model assessment and the driving mechanism of interannual variability" and how this studies fits / differs / is an important step in this continuing development.

**Reply**:

We removed "Model simulation results can be sensitive to model configuration, including the choice of numerical and physical schemes, parameters and grid resolution. We will evaluate a high resolution Arctic Ocean simulation and llucidate the sensitivity of model results to horizontal resolution" from the fourth paragraph, and start directly with the motivation of using high model resolution. This indeed makes this part better. (p2, l32)

The previous work (especially the COREII intercomparison papers) is already mentioned in the third paragraph where we describe the common issues in current Arctic models. In purpose, we tried to avoid lengthy description of FESOM's own applications in the introduction, as there is less intention to review FESOM in this work. Readers can better focus on points of "model resolution" and "Arctic Ocean" without being stopped by paragraphs about unstructured-mesh modelling. Any way, we try to have the short section of Section 5.3, for those who might want to know more about

FESOM and unstructured-mesh modelling in general. We think this is better considering our motivation of the paper, the length of the paper and the readability.

3. Figure 1
This is a good figure and helpful to include for reference. It is also useful to locate the vertical section in figure 8. I suggest the "curves" are better identified as "arrows", since they are orientated markings. In hardcopy, it was difficult to make out the blue arrows and blue text "Transpolar drift" on top of the scalar blue background. I suggest different colours are used to make a clear distinction between the arrows and text vs. the background. Possibly the background scalar depth field could be coloured in green? The arrows outlined in white?
**Reply**: Figure 1 is revised.

4. Figure projections
Figures 2(a) and (b) are compared – and it is good to discuss resolution of the Rossby radius in this way – but they are presented in different projections, which makes a comparison by eye difficult. Moreover, Figure 1 is used for orientation – again this is helpful – but it is oriented differently to 2(b) and the subsequent 4, 6 7, 10, 11, 14 and 15. Please make the orientation consistent, such that Figure 1 can be used to easily identify regions when later output figures are discussed.
**Reply**: The projection of Fig. 1 is changed to the same direction (North Atlantic is at the bottom) as in other figures. We try to keep the previous projection of Fig. 2a because it nicely shows model resolution in a much larger area, and this will not cause any difficulty in understanding the paper content to readers.

5. Definition of HIGH and LOW
The introduction refers to Figure 2 on page 4. Figure 4, on page 5, refers to HIGH, but this is not defined until later in the text on page 6. Please fix so the reader knows what HIGH refers to before it is used in text.
**Reply**: The caption of Fig. 2 is modified as "The horizontal grid size of a mesh with 4.5 km in the Arctic Ocean (referred to as mesh HIGH in this paper)"

6. Eddy resolving considerations
Figure 2 and the connected text make an analysis of where the fixed mesh can resolve mesoscale eddy activity. This quantitative analysis, including the spatial dependence presented in 2(b), is then significantly undermined by the Figure 2 caption comment: "Note that effective model resolution usually is coarser than the grid size due to numerical dissipation." Please change this into a quantitative argument. As it stands the reader really is at a loose end in knowing where the model under consideration is mesoscale eddy resolving. This point is part of the discussion, and the reason for figure 2(b) – and link to / motivation for the increased resolution – so it is important it is completed. Is there a quantitative study of FESOM1.4 (or the underlying discretisation/numerical dissipation present) that can be used to infer the effective model resolution, and where in fact, in this configuration being evaluated here, the model is mesoscale eddy resolving?
**Reply**: Two grid cells per Rossby radius is the boundary of resolution where models may start to resolve mesoscale eddies. We write "Judged by comparing the Rossby radius and grid size (Fig. 2b) and inspecting the simulation result, mesh HIGH is not well eddy resolving in the Arctic Ocean, while it permits eddies in the Eurasian and Canadian Basins." (p6, l6) to be more clear. So far we did not carry out quantitative analysis of effective resolution. By inspecting the model results, we believe that the model permits mesoscale eddies, but does not well resolve them in the Arctic Ocean at 4.5 km resolution. Higher resolution simulations are planned for the future work.

7. Reproducibility and Zenodo archive
The authors have provided a DOI link to a Zenodo archive.
I was able to access the link at https://doi.org/10.5281/zenodo.831484

and download the archive fesom1.4.tar.bz2

After attempting various methods, unfortunately I could not access the contents of the archive.

For example, trying:

tar tvjf fesom1.4.tar.bz2

tar: This does not look like a tar archive

tar: Skipping to next header

tar: Exiting with failure status due to previous errors

Please take a look at the archive. It appears the format is not as expected.

Please provide instructions on how to access the data. Alternatively, and a better solution in my opinion, I suggest uploading files as they are, and not compressed or contained in an archive (Zenodo can handle large single files 50GB) – to avoid problems in opening/uncompressing files.

**Reply**: Bugs in the uploaded files are corrected, and the new DOI is provided in the code and data availability section.

8. Mesh set-up and reproducibility

A large focus of the paper is on the three mesh configurations, LOW, HIGH and high only in the CAA. Sometimes the labels "24 km" and "4.5km" are used for LOW and HIGH in the figures. It could be helpful to keep this consistently to the defined labels LOW and HIGH. For the third mesh configuration with high resolution only in the CAA, various labels are used. "CAA HIGH" appears in figure 10 but is not defined. The graphic uses "CAA 4.5 km". It would be helpful to stick to one label for all throughout. Please define a clear label for this third case, and ideally together with the definitions of the LOW and HIGH cases. It would also be helpful to make it clear in this same place early on that there are 3 different mesh configurations considered here.

**Reply**: We now define the 3 setups in the model description section. In terms of naming, we use LOW, HIGH and HIGH-CAA in the text including figure captions. (p6 l8) In the title of figure panels and figure legends, we try to write the resolution explicitly, in purpose, as many readers will only look at figures first; they will get the model resolution quickly without spending time on looking into the text.

The description of how the resolution varies and in particular how it varies between the three is not described sufficiently. This makes it difficult to reproduce or even compare results. Another flexible mesh model may wish to use 4.5km resolution in the Arctic in the same way to compare and contrast. It is not clear how this would be achieved without significant ambiguity over how resolution is varied, bar the broad, general description "On the second mesh (HIGH) the horizontal resolution is further increased to 4.5 km inside the Arctic Ocean (defined by the Arctic gateways of Bering Strait, CAA, Fram Strait and Barents Sea Opening, Fig. 2a)" and the image in figure 2a (which would be difficult to extract this data from and also includes only half the globe). This paper is evaluating the sensitivity of output to mesh resolution. It is important this is characterised fully. In what way does the resolution vary from 4.5km to 24km in mesh HIGH? The 4.5km region is defined as the Arctic Ocean, closed with reference to the gateways. Yet this description appears incomplete, since in figure 2a it appears higher resolution is applied in the Baltic Sea also? On reproducibility, can the bounds of the 4.5km region used in this case be defined by a mathematical function, in terms of longitude and latitude for this purpose? If not, can shapefile definitions be provided? Other multi-resolution models may not use a triangular mesh, and require accurate description of regions for generation. Please provide sufficient information such that the 3 three mesh configurations can be regenerated – in storage that is persistent and can be reliably depended on, ideally with a DOI. The description of the third configuration is even less well-defined: "It has resolution similar to LOW outside CAA, but the same resolution as in HIGH inside CAA." and leaves the reader questioning why is it not the same resolution as LOW outside the CAA? How is the CAA region defined? How does the resolution vary as you move to outside of the CAA? This is an important study that others will want to compare to. It is important the 3 configurations can be regenerated. Also, ideally there will be sufficient information that others would be able to re-run the same FESOM simulations. The authors advocate a model

intercomparison using high resolution cases. It seems this would be facilitated with sharing as much as possible of the model configurations and model output. Why not make the 3 mesh configurations available on a persistent resource such as Zenodo? It would also benefit a model intercomparison if key outputs from FESOM1.4 here were also made available in this way – for others to contrast and compare.

**Reply**: We agree that storing all the 3 meshes and making them available to others is helpful. And actually all the concerns in this comment can be solved if we do it. Now they are put to the same place as the code with the same DOI. In the paper we stick to what we have and the information is enough for readers who do not want to carry out exactly the same configuration. As the generation of meshes is very complicated with much manual work in between, readers need to download the mesh files and get into the details if they are interested in it for some reason. People who are using different models can generate their own meshes following our mesh data.

9. Spin up and simulation time frames
The spin up part of model runs is repeatedly referred to throughout, but it is not clear how long this is and what forcing data is used. Section 4.2 mentions it takes nearly 30 years for salinity to reach a quasi-equilibrium state. Is this enough for the whole model? In the section on freshwater 5.2, 20 to 30 years is mentioned for a specific sensitivity run exploring the effect of SSS restoring. Please clarify. Page 6, line 29 implies the 3 configurations are run from 1950 to 2009 (with some spin up assumed beforehand, relaxing to a set climatology?). This is supported by figures such as 3, 8 and 9 with time axes over this period.

**Reply:** As mentioned in the model setup section, we start the model simulations from steady ocean, and run them using forcing from 1950 to 2009. The first 20 years are considered as spin up. This is the spinup time for the Arctic basin, but the global ocean needs much longer time to reach equilibrium. We focus on the 60 years period and analyze both the spinup phase and the mean state of the last 20-30 model years. Discussing the spinup phase is very important because the drift taking place in this period, especially the initial surface salinity drift, is the largest (as shown in Wang et al, 2016b). Longer simulations with high resolution will be carried out in future projects.

Figure 4 shows an average from 1970-1999, figure 8 from 1980-1999, figure 6 compares the 70s to 90s, figure 7 is not clear on range considered. Figures 10 and 11 considers different periods 1993-2002, 2003-2007 and 1996-2009, and figure 14 and 15 other ranges. In some cases these choices are justified, but others not. Please make sure it is clear why particular time ranges have been chosen. For example, why in figure 8 miss the contribution from 1970-1979 which is included in figure 4? Indeed why not an average of the entire simulated record over the CORE-II time range?

**Reply**: The choice of analyzed periods is mainly due to two reasons: the availability of observations and published model results; the events/phenomena that should be addressed. The former is related to Figures 4, 6, 10, 11, 14, the latter involves figures 8e,f and 15. The first 20 years are spinup and should be excluded when computing mean and variability, but are addressed when discussing the model spinup behavior. In the figure captions of Figs. 4, 6, 8e,f, 10, 11, 14 we add the reasons for the periods used. In Fig. 7 we show the standard deviation, for which we do not have observations, so we just take the last 30 years for analysis, leaving out the first 30 years; And the main conclusion about the impact of model resolution based on Fig. 7 does not change if we take 10 years more or less. In Fig. 15 we take passive tracers from year 2000, a year after long release of the tracers; but we did not take results from the very last years because the observed warming of Atlantic Water was not well simulated, which might be due to uncertainty in atmospheric forcing used (addressed on page 28, line 12), a topic for future work.

For a paper on model evaluation and more so when different spatial resolutions (and by CFL, time stepping) are involved, one would expect an analysis of model performance with respect to time. This is touched on with a loose reference to 7 days throughput per day. Please include details of time steps used in all three configurations and simulation time per time step on each mesh. If

possible include a scaling plot. It would also be interesting to compare simulation time per time step for each of the model components (e.g. ocean, sea-ice) on each mesh configuration. This information is very important for comparison studies, to guide choices in the development of other models and for users in model choice – possibly even more so here given FESOM2.0 is being developed with a different dynamical core and discretisation choice. On "The deficiency indicates a clear requirement for eddy resolving resolution in the Fram Strait region in order to faithfully simulate the amount and property of AW that enters the Arctic basins through the Fram Strait. In ocean climate simulations, however, it is hardly possible to afford 1 km model resolution in the near future." (page 28, line 3): it would be helpful to be more quantitative here – how much computational effort do the authors believe a 1km version of the model would require? Analysis in Holt et al. (GMD, 2017) implies it will be possible to run global models more routinely down to coastal scales of 1.5km in the next 10 years.

**Reply**:

For climate scale applications, we do not expect modellers can afford 1.5 km in the next 10 years. It will only be practical for process studies and short prediction purposes. That is why we emphasize the importance of improving parameterizations, and Holt et al. (2017) did so too.

We modified the paragraph and added a new table (Table 2) to summarize the computational performance. Thanks for this comment; we noticed an error in the performance estimation (the throughput of HIGH should be about 8, not 7) when we were revising this part.

For us, the model throughput is the most important aspect to consider. We cite a recent paper showing the scaling property of FESOM1.4. We limit our discussion to the comparison of the three grids shown in Table 2, and do not intend to describe model details and do not extend the discussion too much to avoid going beyond the main scope of the paper. (pages 30-31)

The following "Accordingly, efforts on parameterizations are required to improve the simulation of AW circulation in Fram Strait." does not add much. Please add clarification on what parameterisations are required or remove.

**Reply**: changed to "Accordingly, further effort on parameterizing mesoscale eddy effects is required to improve the simulation of AW circulation in the Fram Strait." (p28, l1)

Page 30, line 21:

"However, if the finest grid size is used in narrow straits in FESOM, the model time step and the overall model throughput could be constrained by this grid size." What else is it likely to be constrained by? It seems very likely rather than "could". A breakdown of costs of model components (e.g. ocean, sea-ice, ...), including the cost impact of moving to 4.5km in the Arctic region or just to the CAA – suggested above – would help give a quantitative answer here. As it stands the reader has no feeling for what the next step might be. What is the computational impact of HIGH vs HIGH CAA? – an important aspect of evaluating these model configurations.

**Reply**: The sentence is changed to "However, if the finest grid size is just used in narrow straits, the model time step and the overall model throughput can be constrained by this grid size (the Courant–Friedrichs–Lewy (CFL) constraint)." In terms of computational performance, see our reply above. (p30,l20)

10. Atlantic Water core temperature prediction

Examining Figure 4(a), it appears the LOW model over-estimates the Atlantic Water core temperature (AWCT) compared to PHC climatology. Moving to HIGH this over-estimation increases significantly. Notably warmer waters (of over 1 degree above the PHC climatology) appear to propagate through the Fram Strait into the Arctic Ocean region. In this regard, AWCT temperature *distribution* appears to worsen in the move to using high resolution. Please comment.

**Reply**: This is the feature when the model has higher but not eddy-resolving resolution, shown by Fieg et al. 2010. We have discussed this in the first paragraph of 5.1.b.

How do instantaneous distributions appear? A significant error is seen in HIGH's LFW transport anomaly across the Fram Strait between 1950-1970 (not apparent in LOW) – is this linked? (although Figure 4 contains model results averaged 1970-1999).

**Reply**: The first 20 years are the spinup phase, when the two setups behave very differently. The variability afterwards becomes more similar. So we do not expect a close link between the two phenomena.

On discussing heat content, with both LOW and HIGH integrated content higher, further analysis is dismissed with the following:"Due to inaccuracy in diagnosing heat budget terms (e.g., caused by interpolation) and ignoring heat diffusion in model output, the mismatch between the ocean heat content changing rate and Arctic net heat flux can have the same order of magnitude as this value." (page 26). It is not clear what is meant here. Please reword and improve the explanation. It does not seem satisfactory that the increased heat content cannot be explained. "Our model results show that the magnitude of the AW temperature is not lower in LOW" (page 26) – is this shown in Figure 5? Is this the maximum or depth integrated magnitude of AW temperature?

**Reply**: The first few sentences of this paragraph are removed to avoid misleading. In our practice we found that we cannot close the heat budget for the Arctic ocean domain: summing up all heat fluxes, we get the net heat flux into the Arctic Ocean; there is misfit between this net heat flux and the total heat content change in the Arctic Ocean. The model conserves tracers, so we believe the misfit is mainly due to interpolation errors when we calculate heat transports during offline analysis of the model results, although neglecting diffusion of temperature across gateways in the analysis can also contribute to the misfit. The misfit is small, compared to individual components of heat source/sink fluxes (ocean heat fluxes through different gateways and different components of atmospheric heat fluxes), but it is as large as the heat flux source required to explain the difference of heat content between the two simulations. Therefore, we analyze passive tracers to help us to understand the difference of AW water masses. The paragraph is modified to make it clearer.

11. Over-estimation of liquid freshwater content
Both LOW and HIGH over-estimate the liquid freshwater (FW) content. This is noted on page 18, line 11. What are the reasons for this over-estimation? Is it expected this can be further reduced by further increasing resolution?

**Reply**: We do not expect that the reason is model resolution. Discussion on this was given in section 5.2.a.

"The variety of FW content distributions simulated in different ocean models shown by Wang et al. (2016b) presumably can be partly attributed to different model representations of the CAA region." – this is could be true, but this does not add significantly. Can it be backed up by other studies? What are the different representations? How do they relate to the HIGH CAA configuration here? Do other representations – that are most closely related to the one here – show similar changes/improvements/indications? Some of this is discussed much later on in 5.3 and 6. It appears a key consideration here given focus on resolution and representation and might be good to have its own section?

**Reply**: We used the HIGH-CAA experiment to show that the representation of CAA region can influence the FW content spatial pattern (in section 4.2). It is based on this finding that we speculate that the variety of FW content spatial pattern shown by Wang et al. 2016b can be partly attributed to the CAA representation in coarse models (in section 4.3). In that study, it was shown that climate ocean models have very different width and resolution in the channels, which can certainly produce very uncertain ocean transports. The finding that better resolving CAA transport can impact also the Arctic basin FW distribution and the fact that currently CAA is not resolved in ocean climate models are the major information that we want to address. Each model development group has the task: Inspect the ocean transports through main ocean gateways.

12. Passive tracer implementation

Figure 11 includes the comments "Note that the passive tracers were set to zero south of Fram and Davis Straits. The passive tracers are averaged over the upper 100 m." These are details better included inline in the text. This raises additional questions. For the tracer φ, the above implies:

$$\frac{\partial}{\partial t}\int_V \varphi \, dV = 0 \quad (1)$$

So tracer total is not conserved? Please elaborate on why this is done and its implications. What is the reason for averaging the tracer over the top 100m? Is this type of analysis done elsewhere, under the same conditions?

**Reply**: We did not manually change passive tracers after they enter the Arctic Ocean in the model simulations. We just did not plot them in the figure for the paper. The caption is changed to avoid misleading. As we are discussing the surface water (the FW), we show the passive tracers in the upper ocean.

13. FW variability
Referring to these sentences in section 4.4:
–
"Compared to the period of 1980-2000, the mean Arctic FW content averaged over the 2000s has increased by about 4500 km3 based on observations (Polyakov et al., 2013b; Haine et al., 2015), while the increase is only about 1700 km3 in our two simulations (Table 1)."
–
"On average the 13 CORE-II models analyzed in Wang et al. (2016b) underestimated the observed upward trend also by half." Observations show mean Arctic FW content has increased nearly 3 times that seen in the model simulations presented. Why is this? The paper highlights that on average CORE-II models underestimate by a half. These models have a significantly lower resolution than the HIGH case considered here, yet the above implies this higher-resolution simulation performs more poorly than the average. Why is this?

**Reply**: the first two paragraphs discussed two "different diagnostics". The first is about the "difference of FW content" between two chosen periods which have been considered by Haine et al., 2015. This diagnostic was not analyzed in Wang et al. 2016b. The second diagnostic is the "linear trend of FW content" computed from 1996 to 2009, which we compare to the estimate based on observations (Haine et al, 2015 and Polyakov et al. 2013b) and the analysis of Wang et al. 2016b. The linear trend is slightly higher and closer to the observational estimate in HIGH than in LOW as represented in the second paragraph. Anyway, the difference between HIGH and LOW in terms of the trend of FW content is not very significant. This implies that other model parameters may play large roles for this diagnostic, and we tried to address this point in general including the very last paragraph of the paper (see also the reply to another comment about the very last paragraph above).

The paragraph on temporal variation (page 22, line 7) ends with "Simulation HIGH consistently obtains positive changes in the Eurasian Basin, but with a larger magnitude. It has negative values north of Greenland, which is not present in the observation."
which appear unexplained. Can you suggest why negative changes are seen north of Greenland? and the positive change in the Eurasian Basin?

**Reply**: The FW content in the Eurasian Basin increases between the two periods in the model, consistent to the observation. The background of FW increase in the Arctic Ocean was given in the introduction subsection (4.1). For the change of FW content in the small region north of Greenland, the model shows local decrease. Too many reasons can cause the difference of the model from the observations, including the pathway of liquid FW (of different sources) and the local sea ice condition, and uncertainties due to very sparse observations as well. But such speculation is too general and will not give more information. Based on the fact that the model largely captures the spatial pattern of FW increase, and we just pointed out the place of the clear difference from observations without much general speculation. We add "Further efforts are required to understand the reason." (p22, l9)

On "The interannual variability of FW transport through the Arctic gateways shows large similarity between the two simulations (Fig. 13a-c)." It does not seem correct to characterise this as a large similarity. There is some agreement in the anomaly trend, but also large deviations. Fig. 13(a) shows HIGH has a very significant deviation in the first two decades 1950-1970 of the six decades considered. The magnitude of the deviation is of the order of the max change seen, so arguably significant. Why the deviation? Is it linked to the excessively warmer waters see entering through the Fram Strait in the model output?

**Reply**: As mentioned in one of the above comments, the first 20 years are spinup phase, when the FW transports behave differently in the two simulations especially for the Fram Strait. Afterwards, the correlation is high (Table 1). The period 1980-2009 is considered in the Table. To be clear, in the text we change to "The interannual variability of FW transport through the Arctic gateways shows large similarity between the two simulations *after the spin-up phase* (Fig. 13a-c). The correlation coefficients between the FW transports from the two simulations are similar at the Davis and Fram Straits (0.75 and 0.78, respectively *for the period of 1980--2009*, Table 1). (p23, l1)

14. Outcomes
"Instead, we often modify the geometry of the CAA channels to allow adequate CAA throughflow. Such model adjustment, however, is not trivial as shown by the large model spread in CAA FW transports among the ocean climate models analyzed in Wang et al. (2016b)." (Page 30, line 27). This can be considered yet another parameterisation and the same approaches made to analyse its impact. "When developing global climate models, the modeling groups certainly need more efforts to better adjust the CAA representation. Besides, maintaining high resolution measurements of ocean transports is of great importance for model development too." (Page 30, line 29). This does not add anything significant. Can the authors suggest a solution to CAA representation/parameterisation?

**Reply**: CAA representation means how to adjust the geometry and grids in the CAA channels. In the work of Wang et al. 2016b, it was found that model development groups often treat the narrow straits without examining the results, and people often forget who did the mesh and how it was done. Therefore, our purpose here is to repeat the importance of adequate treatment of the straits and warn model developers to take care. We cannot suggest solutions beyond this, which can be model dependent.

"Most of the models analyzed in past CORE-II model intercomparison studies have relatively coarse resolution. For developing our unstructured-mesh model system with regional focus, it would be helpful to communicate experience with the large structured-mesh model community, for example, in future high resolution ocean climate model intercomparison projects." (Page 31, line 1). This could be better worded. Do you mean "Most of the test cases/configurations in past CORE-II..."? – and you are suggesting a suite of higher resolution intercomparison test cases?

**Rely**: we modify the last sentence to " .... (for example, through the future CMIP projects where increasing model resolution will be pursued (Haarsma et al. 2016))." (p31 ,l 11)

15. Other small points
(a) Last line of the abstract sounds disjoint, given the paper concerns increased resolution. Maybe something like: "Along with increased resolution, we additionally discuss other issues that could benefit from development to help increase accuracy in the region, including the improvement of parameterizations, for example."

**Reply**: changed to "... appear not to be very sensitive to the increase in resolution employed here. By highlighting the issues that are independent on model resolution, we address that other efforts including the improvement of parameterizations are still required." (the last sentence of the abstract)

(b) Lars Smedsrud et al. have recently published (January 2017 – you cite as "Smedsrud et al., 2017" ) on Fram Strait sea ice export – can this help update the 11+ year old figure and error in Table 1? "Fram Strait sea ice export variability and September Arctic sea ice extent over the last 80 years" The Cryosphere, doi:10.5194/tc-11-65-2017. There appears to be some large deviations in Fram Strait fluxes here, comparing FESOM model output and observations.
**Reply**: Uncertainties in observations come mainly from ice thickness estimate. The Smedsrud paper does not intend to solve this issue, but rather focuses on "*area* flux".

(c) Page 24, line 21: Part of a general request for more quantitative statements: "Note that much higher model resolution is required in order to simulate sea ice leads with realistic width." What resolution is required? Is it therefore expected that increase in sea-ice model resolution which see no advantage until this is reached?
**Reply**: changed to "... because they are typically narrower than 1 km in reality (Tschudi et al., 2002)." At least when such width can be properly resolved, the related physical processes can be directly simulated. Otherwise parameterizations are required. (p24,l21)

**Technical corrections**
1. Use of the definite article "the" appears to have been skipped in multiple places throughout the paper. e.g. Page 10, line 6: "The maximum temperature in Eurasian Basin in simulation" , Page 11, line 8: "correct circulation direction in Canadian Basin".
Reply: we corrected these place and tried to proofread the text.

2. Page 1, line 14:
"including improving parameterizations" , better: "including the improvement of parameterizations".
Reply: changed. (p1, l14)

3. Page 1, line 22: "deep water formation regions"
better worded as "regions where deep water is formed"?
Reply: changed (p1, l22)

4. Page 2, line 16:
"lower latitudes ocean and climate", change to "lower latitude ocean and climate".
Reply: changed (p2, l16)

5. Page 2, line 16:
"societal" is better here than "social".
Reply: changed (p2, l16)

6. Page 2, line 16:
"that remains under debate" better.
Reply: changed (p2, l16)

7. Please check capitalisation in references, e.g.
– Wekerle et al. (2013) should contain "Canadian Arctic Archipelago".
– Schauer et al (2002) "Amundsen" and "Makarov".
– "Arctic" , "Fram" and "Barents" in Maslowski et al. (2004).
– …
Reply: changed

8. Page 25, line 3:
"(e.g., Smedsrud" comma not needed – please check throughout.

Reply: this depends on the journal requirement. We leave this for the production editor to decide and change.

9. Figure 14(f) is not labelled "(f)". Also, the coastline is not identified like the others
(a)–(e).
Reply: changed

10. Figure 2 caption:
"Values larger than one indicate mesoscale eddy resolving." the meaning is understood, but please make into a sentence.
Reply: changed to "With resolution finer than two grid cells per Rossby radius models may start to resolve mesoscale eddies...."

11. Page 5, line 3:
"It work with" , change to "It works with".
Reply: changed (p4, l42)

12. Page 5, line 15:
"shows *the* least".
Reply: these sentences are removed now.

13. Page 6, line 21:
"one over the ocean column"
– is this one over the ocean depth? water column thickness?
Reply: changed to "in the whole ocean column". (p6 ,l1)

14. Pages 4 and 6: Please avoid repetition of the
"Time frame of 7 model years per day".
Reply: the second-time phrase is removed (page 6)

15. Min and max values are missing in many colour bars – e.g. Figures 2b, 3a, 3c, 7a, 7b, 8a-e, 10, ... – This is very helpful for future studies comparing to this work. Please ensure all min and maxes are labelled.
Reply: Revised.

16. Some axis do not have end values – e.g. time axes in Figure 9. Is this data until 2009? Please make it clear where the axis ends (mark with year, or make it a decade to 2010). Same with Fig 3, 12, 13, …
Reply: the shown integration period is added in the figure captions.

17. Use of the definite article "the" appears inconsistent – e.g.
"towards the Fram Strait" on page 7 (twice), but "toward Fram Strait" on page 8.
Reply: missing "the" is added.

18. Spelling of "elucidate" page 2, line 34.
Reply: sentence removed.

19.
"Dupont, F., and others (2015)"
– please elucidate on the "others".
Reply: corrected

20.

"CAA" unnecessarily redefined in figure 10.
Reply: For readers who only read figures, we prefer to explain abbreviation in the figure captions.

21. Possibly move the reference to Holloway et al. (2007) on page 11 up to the introduction of topostrophy – so readers know where to find its definition.
Reply: the reference is added after "topostrophy". (p11, l1)

22. Page 11, line 10:
"Indeed, the Arctic Ocean hydrography obtained on mesh LOW was found to be one of the well simulated when comparing the state-of-the-art ocean climate models (Ilicak et al., 2016)"
"one of the well simulated" does not read well and the sentence is verbose. Last part better rephrased as "when compared to the suite of state-of-the-art climate models in Ilicak et al. (2016)"?
Reply: changed to "Indeed, the Arctic Ocean hydrography obtained on mesh LOW is well simulated when compared to the suite of state-of-the-art ocean climate models analyzed in Ilicak et al. (2016)." (p11,l5)

23. Page 12, line 10:
"between the two models" or "between models".
Reply: changed

24. Page 18. line 2: Please refer directly to the section number rather than this loose reference.
Reply: "Section 4.2" is added. (p20, l6)

25. Page 24, line 11:
"in *the* 2000s".
Reply: added (p24, l6)

26. Page 30, line 4:
"without necessity" , please reword – "without *the* necessity"?
Reply: added (p30, l3)

27. Page 33, line 1: Space in URL.
Reply: space removed (p33, l16)

Best regards
the authors

---

## Referee Report (RR1)

**Review of gmd-2017-136, version 2**

Recommendation: minor revision

**General comments**

The authors have updated the manuscript in light of the comments provided. In particular:

1. It is now possible to access source files for the FESOM 1.4 code used in the simulations. The error in the tar archive previously provided has been fixed.

2. Figure 1 has been fixed. Agree that figure 2(a) is good as is (except for a question over the higher resolution seen in the Baltic Sea – see below).

3. Labelling of the 3 cases has been made consistent and introduced earlier, before first use.

4. Meshes are provided, enabling setups to be run (although more could be done to improve ability to make comparisons in the future – see below).

5. Model performance table added, which is very helpful, and essential for future comparisons.

6. Explanation in 5.1(a) improved.

7. FW variability and over-estimation of LFW clarified.

8. Some of the highlighted repetition has been removed.

As highlighted in the initial review, the research is timely, investigating an interesting question of model spatial resolution in the Arctic. This is directly relevant to current developments in coupled climate models, as flexible resolution models enter challenging inter comparison studies such as CORE-II. Whilst it is somewhat unsatisfactory to take a single mesh choice, run the coupled model in this configuration and analyse the output there is still the potential to learn significantly from this type of exercise.

There are, however, still significant concerns over the verbosity and unnecessary length of the paper. The length has not changed. It still stands at 41 pages. A more concise report would help make the paper accessible. There are still many sentences that are too general, lack specifics or quantitative backing. I disagree that so much repetition is required, which acts to dilute the key messages of the paper. Some of the structure has been changed, but this could be further improved also. On these first two points I defer to the topical editor as to what level of repetition and verbosity is appropriate for GMD.

Aside from this, there are a few other remaining issues to address regarding a discrepancy between text and figure on resolution in the Baltic, and why passive tracers are not fully plotted (and again the text here does not appear to describe what appears in the figure). See below for further details.

**Specific comments**

1. *Focus.* Requests were made to improve lucidity, removing general statements that do not add to the report. Focusing and reducing verbosity would make the paper more accessible. Whilst some repetition and verbosity has been reduced, as suggested, the authors have chosen to keep a large proportion of the content which appears superfluous, and does not add to the paper. The paper is still at 41 pages in length, despite these requests to shorten and make concise. There are still many sentences that are too general, lack specifics or quantitative backing. e.g. referring to 'common issues" with no specifics that are not helpful. Another:

"with a very reasonable thickness" – how good is the AW modelled? A quantitative statement gives a basis for others to compare to (and improve upon) in the future. Further examples were highlighted in the initial review.

On the advection scheme: Discussion of the FCT scheme still remains. Nothing profound, since it would be unusual to use an advection scheme which does not reduce numerical smoothing as resolution is increased! The discussion here again seems unnecessary. It is accepted higher resolution gives more accurate representation of mixing processes. This is not specific to the FCT scheme mentioned. Moreover, "numerical mixing" is possibly better termed "numerical smoothing" or "diffusive solution"?

2. *Structure.* The reasons to apply high resolution has been repositioned earlier as suggested, which helps lead a reader through the motivation.

   Points raised on section 5.3 and a request for context within FESOM efforts (since the 4.5km mesh has been used elsewhere in another publication) have not been adopted.

   This would be helpful for not only those following FESOM, but unstructured mesh model contributions to CORE-II.

3. *Figure 1.* Figure 1 has been fixed. Agree that figure 2(a) is good as is.

4. *Figure projections.* Fixed, such that comparisons are easy to make by eye.

5. *Definition of HIGH and LOW.* Labelling of the 3 cases has been made consistent and introduced earlier, before first use. Clear definition of the CAA case also now made.

6. *Eddy resolving considerations.* A quantitative statement has now been included.

7. *Reproducibility and Zenodo archive.* It is now possible to access source files for the FESOM 1.4 code used in the simulations. The error in the tar archive previously provided has been fixed.

   A suggestion was also made to provide the model output to facilitate future comparisons. This again would help model intercomparison efforts.

8. *Mesh set-up and reproducibility.* Meshes have now been provided.

   This is good, but a more general prescription of mesh generation, as suggested, would provide a more rigorous foundation for other comparative studies. These provided raw mesh descriptions can only be used by a limited number of other models that are similar to FESOM1.4. For example, it would be useful to include a more thorough description of how mesh size changes over Long-Lat space to enable generation in general – i.e. the element size metric. Currently, others will need to analyse the meshes to infer how this was chosen.

   This again would help improve possibilities for model intercomparisons.

   On meshes for intercomparison studies, the authors comment: "...and people often forget who did the mesh and how it was done." Why not provide a more complete description and lead the solution to this prevalent problem in the field?

   Also, the point regarding the higher resolution that seems to appear in the Baltic Sea in figure 2(a) has not been addressed.

9. *Spin up and simulation time frames.* This has been explained, and justification for selected time frames usefully included.

10. *Atlantic Water core temperature prediction.* Points raised have been clarified in text.

11. *Over-estimation of liquid freshwater content.* Explanations accepted. On the second reply discussing HIGH-CAA, the point was to make the context now provided in the response clear in the paper. This is a key point but is somewhat lost in the paper.

12. *Passive tracer implementation.* Before, the implication was that it was set to zero in the simulation, so it is helpful this has been clarified – although to avoid ambiguity, why not just say the passive tracer is not plotted below the Fram and Davis Straits?

    This does lead the reader to wonder why the passive tracer is not plotted in these regions. Why not plot the tracer over the entire domain?

    Referring to the location of the Davis Strait marked on Figure 1, it does not look to be true that the passive tracer is not plotted there in Figure 11 (a) and (b), both (middle) and (right).

    Do the authors mean it is not plotted in the region bounded by the Fram Strait and the Barents Sea Opening? Please correct. Again, why is not plotted there?

13. *FW variability.* Thank you for the explanations and clarifications in the paper.

14. *Outcomes.* Again, thank you for the explanations and clarifications in the paper.

15. *Other small points.* Explanations and text changes help clarify.

**Technical corrections**

All complete – good.

A few other minor points seen in the updated version:

1. Page 1, line 13: "independent on" better replaced by "independent to"?

2. Page 14, lines 6-9: Two sentences in a single parenthesis. Better to remove and start a fresh sentence: "content (Defined " → "content. This is defined "

3. References: JGR Oceans referred to as: "J. Geophys. Res. - Oceans", "Journal of Geophysical Research-oceans", "J. Geophys. Res. Oceans"

4. Page 37: "Mcwilliams" → "McWilliams"

---

## Author Response (AR2)

Dear Editor,

Thank you for your help during the review process.

The remaining concern was on the length of the paper. As the paper is organized, partly, as a report for FESOM users starting with Arctic simulations, we did in purpose describe the background of the Arctic Ocean topics that we want to address. And then in the main result sections we follow the storyline and showed details of our model results. This leads to a relatively long paper. However, we think such details (background, model results and discussions about future work) are necessary and very useful for some new model users, who are the potential readers of the paper. And this strategy is not against the basic idea of the GMD policy. For other few minor comments, we did the corrections or replied accordingly in the reply letter.

Sincerely
the authors

Dear reviewer,

Thank you very much for your time spent on improving our manuscript.

The remaining concern was on the length of the paper. As the paper is organized, partly, as a report for FESOM users starting with Arctic simulations, we did in purpose describe the background of the Arctic Ocean topics that we want to address. And then in the main result sections we follow the storyline and showed details of our model results. We think such details (background, model results and discussions about future work) are necessary and useful for some new model users, who are the potential readers of the paper. This strategy is not against the basic idea of the GMD policy.

The reply to the few remaining comments:

1. Focus. Whilst some repetition and verbosity has been reduced, as suggested, the authors have chosen to keep a large proportion of the content which appears superfluous, and does not add to the paper.
**Reply**: see reply to the general comment above.

For the few specific examples you raised:

There are still many sentences that are too general, lack specifics or quantitative backing. e.g. referring to 'common issues" with no specifics that are not helpful.
**Reply:** the 'common issues' have been described in the Introduction Section. To remind this, now we add „as mentioned in the introduction section" in the first paragraph of the Conclusion Section when we come to the „common issues" again. (p31, l20)

Another: "with a very reasonable thickness" – how good is the AW modeled? A quantitative statement gives a basis for others to compare to (and improve upon) in the future. Further examples were highlighted in the initial review.
**Reply:** In the Conclusion Section, this sentence summarizes Section and related figures, where one reads quantitative comparison. Now we add the reference to the Figure in this summary sentence. (p32, l14)

On the advection scheme: Discussion of the FCT scheme still remains. Nothing profound, since it would be unusual to use an advection scheme which does not reduce numerical smoothing as resolution is increased! The discussion here again seems unnecessary. It is accepted higher resolution gives more accurate representation of mixing processes. This is not specific to the FCT scheme mentioned. Moreover, "numerical mixing" is possibly better termed "numerical smoothing" or "diffusive solution"?
**Reply**: We did remove the discussion about FCT in the model description section already. Only one sentence in the Conclusion Section was left; now we removed this sentence too.
Numerical mixing is an acceptable phrase in ocean modeling.

2. a request for context within FESOM efforts (since the 4.5km mesh has been used elsewhere in another publication) have not been adopted. This would be helpful for not only those following FESOM, but unstructured mesh model contributions to CORE-II.
**Reply**: We believe that it is better to keep the discussion about unstructured meshes in the discussion section as it is now. In the general introduction section, we already brought up FESOM enough through the review of Arctic simulations in past studies. We prefer to discuss the Arctic Ocean simulation in a context of „ocean general circulation models" and try to just emphasize the special points about unstructured meshes afterwards (section 5.3). We are not developing a new model code or version here, and  rather want to present the paper to general Arctic modelers too.

8. Mesh set-up and reproducibility. Meshes have now been provided. This is good, but a more general prescription of mesh generation, as suggested, would provide a more rigorous foundation for other comparative studies.

**Reply**: If one wants to generate a new unstructured mesh following the exact resolution we use, it is necessary to look at the mesh we provided, which tells directly the grid size. This is the best practice. We notice that mesh generation itself is a hard and practical task, because it is very generator-specific, and beyond the scope that this single paper can cover.

Also, the point regarding the higher resolution that seems to appear in the Baltic Sea in figure 2(a) has not been addressed.

**Reply**: We examined it now.This feature is related to the generator setting: the small region/corner in the Baltic Sea is close to the Arctic in distance, so it gets a bit higher resolution. As the mesh files are provided, one can see it from the mesh in case of reproducing a new mesh. Note: this corner with increased resolution on the mesh, however, does not influence the general circulation we are studying.

12. Passive tracer implementation.

**Reply**: The focus is on the Arctic simulation, so we choose to only show the region that is relevant or where we want to address it. Otherwise we prefer to remove it from the plots to avoid too much distraction. We did it in an easy way by specifying a lon/lat range, so it does not remove all the color outside the Arctic Ocean following the exact complicated Arctic boundary. But it is clear to readers that we only focus on the Arctic Ocean in this work.

A few other minor points
1. Page 1, line 13: "independent on" better replaced by "independent to"?
Reply: *changed.*
2. Page 14, lines 6-9: Two sentences in a single parenthesis. Better to remove and start a fresh sentence: "content (Defined " → "content. This is defined "
Reply: *changed*
3. References: JGR Oceans referred to as: "J. Geophys. Res. - Oceans", "Journal of Geophysical Research-oceans", "J. Geophys. Res. Oceans"
Reply: *changed to the same abbreviation in the references list*
4. Page 37: "Mcwilliams" → "McWilliams"
Reply: *changed*

Sincerely
the authors